# *RET* rearrangements are actionable alterations in breast cancer

Bhavna S. Paratala[1,2], Jon H. Chung [3], Casey B. Williams[4], Bahar Yilmazel[3], Whitney Petrosky[1,2], Kirstin Williams[4], Alexa B. Schrock[3], Laurie M. Gay[3], Ellen Lee[5], Sonia C. Dolfi[1,2], Kien Pham[6], Stephanie Lin[1,2], Ming Yao[1,2], Atul Kulkarni[1,2], Frances DiClemente[1,2], Chen Liu[6], Lorna Rodriguez-Rodriguez[2,7], Shridar Ganesan[1,2], Jeffrey S. Ross[3], Siraj M. Ali[3], Brian Leyland-Jones[4] & Kim M. Hirshfield[1,2]

Fusions involving the oncogenic gene *RET* have been observed in thyroid and lung cancers. Here we report *RET* gene alterations, including amplification, missense mutations, known fusions, novel fusions, and rearrangements in breast cancer. Their frequency, oncogenic potential, and actionability in breast cancer are described. Two out of eight *RET* fusions (*NCOA4-RET* and a novel *RASGEF1A-RET* fusion) and *RET* amplification were functionally characterized and shown to activate RET kinase and drive signaling through MAPK and PI3K pathways. These fusions and *RET* amplification can induce transformation of non-tumorigenic cells, support xenograft tumor formation, and render sensitivity to RET inhibition. An index case of metastatic breast cancer progressing on HER2-targeted therapy was found to have the *NCOA4-RET* fusion. Subsequent treatment with the RET inhibitor cabozantinib led to a rapid clinical and radiographic response. *RET* alterations, identified by genomic profiling, are promising therapeutic targets and are present in a subset of breast cancers.

[1] Department of Medicine, Division of Medical Oncology, Rutgers Cancer Institute of New Jersey, New Brunswick, NJ 08901, USA. [2] Rutgers University, Piscataway, NJ 08854, USA. [3] Foundation Medicine, Cambridge, MA 02139, USA. [4] Avera Cancer Institute Center for Precision Oncology, Sioux Falls, SD 57105, USA. [5] University Radiology Group, New Brunswick, NJ 08901, USA. [6] Department of Pathology and Laboratory Medicine, Rutgers New Jersey Medical School and Rutgers Robert Wood Johnson Medical School, Newark, NJ 07103, USA. [7] Department of Obstetrics and Gynecology, Division of Gynecologic Oncology, Rutgers Cancer Institute of New Jersey, New Brunswick, NJ 08901, USA. Correspondence and requests for materials should be addressed to B.L-J. (email: brian.leylandjones@avera.org) or to K.M.H. (email: kim.hirshfield@merck.com)

Oncogenic *RET* (rearranged during transfection) gene fusions have been described in 6.8% of papillary thyroid (PTC) and 1–2% of non-small cell lung cancers (NSCLC)[1,2]. Fusion-related structural alterations of RET, a transmembrane receptor tyrosine kinase, lead to ligand-independent dimerization and constitutive kinase activation in turn driving oncogenic signaling cascades and increased cell survival and proliferation[3]. Most RET fusions lack the N-terminal and transmembrane domains leading to aberrant localization of the fusion product, thus avoiding normal intracellular trafficking and degradation[4]. Currently, multi-kinase inhibitors with activity against RET are FDA approved for thyroid cancers, and clinical trials are investigating their use in targeting *RET* fusions in lung and other solid cancers[5,6].

Studies have demonstrated elevated expression of RET in ER+, HER2+, and a subset of ER− breast cancers[7,8]. RET overexpression in ER+ breast cancer was associated with resistance to tamoxifen and aromatase inhibitors[9–11]. In contrast, reduced RET expression, resulting from a *RET* polymorph, correlated with improved overall survival of patients with ER+ breast cancer[12]. Combining the aromatase inhibitor, letrozole, with a RET inhibitor has demonstrated improved efficacy over letrozole alone in preclinical models[13] and clinical trials are evaluating the use of RET inhibitors to enhance sensitivity and reduce resistance to hormonal therapies in breast cancers (clinicaltrials.gov).

Although the role of RET expression in ER+ breast cancer has been under investigation, a comprehensive analysis of the presence and frequency of recurring *RET* genomic alterations, particularly *RET* rearrangements in breast cancer has not been reported. Unlike targeted or whole exome sequencing approaches, intron-capture-based approaches that involve baiting of targeted intronic regions can identify rearrangements, including translocations, inversions, tandem duplications or small deletions. By high depth sequencing of hotspot introns that precede the kinase domain-coding exons, breakpoints of rearrangements affecting these regions can be identified. Similar to *RET* mutations, *RET* rearrangements in breast cancer may serve as predictive biomarkers and be therapeutically actionable using small molecule kinase inhibitors[14–16].

Using targeted genomic profiling with hybrid capture that includes analysis of introns 9, 10, and 11 of *RET*, high-level analysis of genomic alterations of *RET* in a variety of breast cancer subtypes are presented here. Based on initial detection in two index cases with *RET* fusions (*RASGEF1A-RET* and *NCOA4-RET*), 9693 breast cancers were evaluated for the presence of *RET* rearrangements as well as missense mutations and copy number changes. Novel rearrangements and previously reported fusions in *RET* were identified in this comprehensive cohort of advanced breast cancer patients. The presence of a spectrum of structural events and copy number alterations involving *RET* is demonstrated here, including co-events, functionality, and therapeutic actionability of rearrangements and amplifications using in vitro and in vivo models. Lastly, clinical benefit was observed in the index case harboring *NCOA4-RET* after receiving the RET inhibitor cabozantinib.

## Results

**Genomic profiling identifies recurrent RET alterations**. The landscape of *RET* genomic alterations in 9693 breast cancer samples was assessed as part of hybrid capture-based next-generation sequencing of up to 405 cancer-related genes including select introns of up to 25 genes (gene panels, Supplementary Data 1). Samples were sequenced to a high uniform depth of coverage (median exon coverage, 637×). Median patient age was 54 years (range, 20−88) (Table 1) and all samples were from

female patients. Tissue for genomic profiling was obtained from the breast for 3859 patients (40%) and from metastatic sites for 5834 patients (60%).

*RET* genomic alterations were observed in 1.2% (121/9693) of breast cancer cases. With one case harboring two *RET* alterations (*RET* amplification and *RET* rearrangement), a total of 122 *RET* genomic alterations were identified including 16 rearrangements, 25 missense mutations, and 81 amplifications (median copy number = 8, copy number range = 6–21) (Fig. 1a, Table 1, Supplementary Data 2). ER status, based on immunohistochemistry, was available for 81% (98/121) of *RET* altered cases and HER2 *(ERBB2* amplification) status, based on genomic profiling, was available for all cases (Fig. 1a, Table 1). *RET* genomic alterations were detected across all breast cancer subtypes although a majority were ER− (65%) or *ERBB2* nonamplified (82%). In comparison to cases with other types of *RET* genomic alterations (Fig. 1a, Table 1), the subset of cases with *RET* missense mutations was more frequently ER+ (71%) ($p = 0.0002$, Fisher's exact test, two-tailed; Table 1). *RET* rearrangements were more frequently ER− (75%) ($p = 0.1615$, Fisher's exact test, two-tailed; Table 1), as were *RET* amplifications (75%) ($p = 0.0077$, Fisher's exact test, two-tailed; Table 1). For the 121 *RET* altered cases, the most frequent altered genes that co-occurred with *RET* were *TP53* (80%), *MYC* (32%), *PIK3CA* (26%), *ERBB2* (20%), *MCL1* (20%), and *PTEN* (17%) (Supplementary Table 1).

Rearrangements were identified by hybrid capture using probes for selected hotspot introns (introns 9–11) and all exons of *RET*. Based on detection of breakpoints, all rearrangements retaining intact sequences for *RET* kinase domain-coding exons (12–19), with or without N-terminal fusion partner genes were included for the analysis. Out of the 16 *RET* rearrangements identified, 8 were defined as activating fusions, and the other 8 as uncharacterized rearrangements. Of the eight activating fusion events predicted to contain the kinase domain of RET, seven cases were either *CCDC6-RET* ($n = 6$) or *NCOA4-RET* ($n = 1$) that have been previously characterized as oncogenic[17–19] and are recurrent in NSCLC[20] and PTC (Fig. 1a, b); one case harbored a novel *RASGEF1A-RET* fusion subsequently characterized here as functional and activating.

Of the total eight unique uncharacterized rearrangements identified, novel gene partners, duplications, and truncations were observed (Fig. 1c). One case harbored an *RET-RASGEF1A* rearrangement. Although this rearrangement did not include the exons encoding the kinase domain, the similarity of the genomic breakpoints to the *RASGEF1A-RET* fusion (Fig. 1b) suggests the potential for a reciprocal rearrangement event that was not detected but could potentially lead to an activating fusion. One case harbored a *ZNF485-RET* rearrangement where ZNF485 is a putative novel fusion partner; four cases harbored full-length *RET* (including the 3′ untranslated region or UTR) followed by tandem duplication of the *RET* 3′ exons including exons 12–19 that encode the kinase domain; two cases harbored rearrangements with a breakpoint at *RET* intron 11 with exons encoding the kinase domain juxtaposed with intergenic space.

Missense mutations in *RET* were detected in 25 cases (Fig. 1d) and the majority were characterized as activating (64%, 16/25). Observed recurrent, *RET* activating point mutations included the E511K extracellular domain mutation ($n = 5$)[21] and V804M kinase domain mutation ($n = 4$)[3]; other activating mutations that were observed include extracellular domain mutation C611R, C620F, L633V, C634R, C634F, T636M, and the kinase domain M918T[3,22]. The remaining missense mutations have been previously described as somatic in cancer but have not been characterized[23]. *RET* missense mutations are known to be causally associated with MEN2 syndrome (multiple endocrine neoplasia type 2). Using a computational method, missense

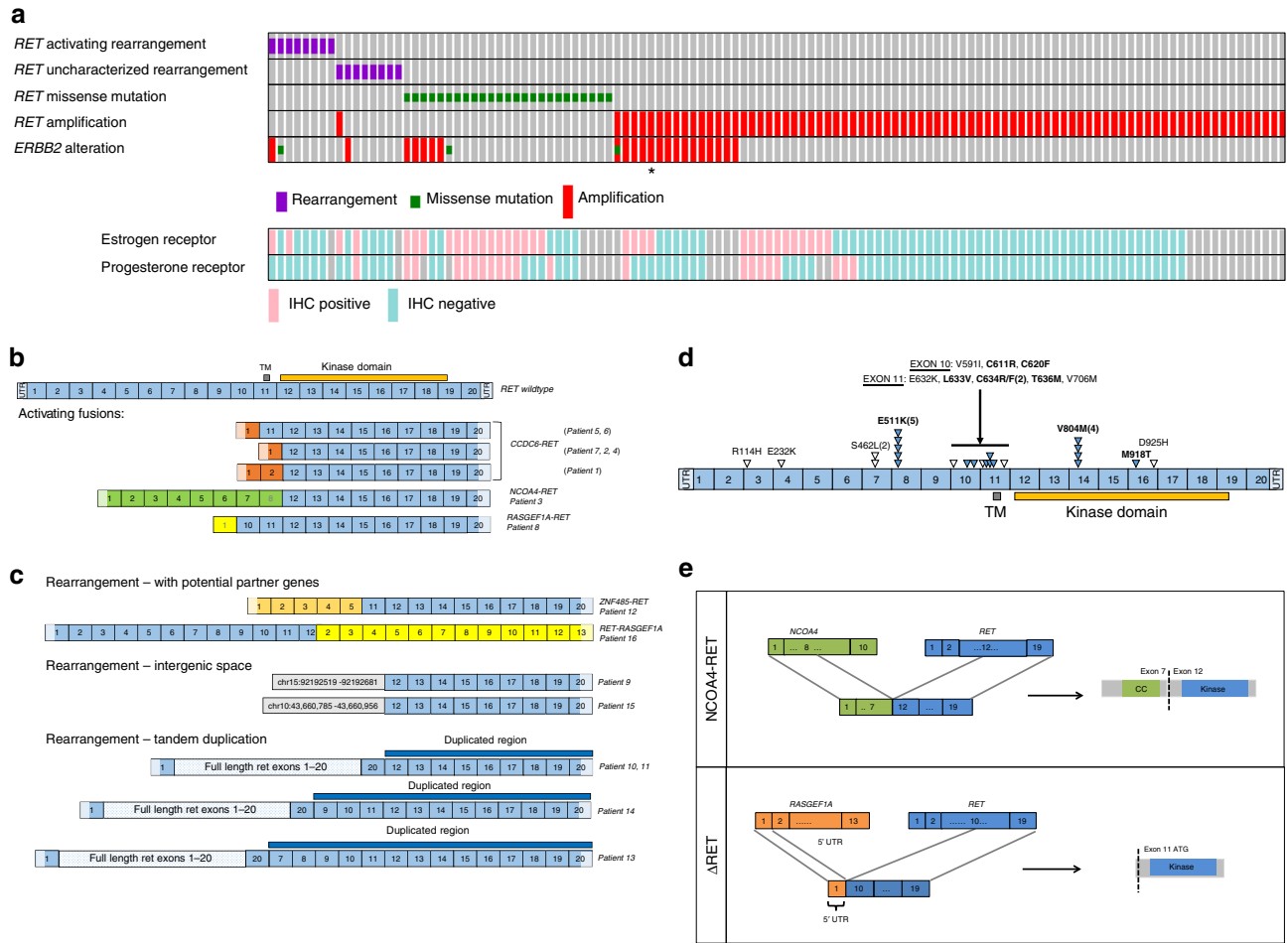

**Fig. 1** Recurrent *RET* rearrangements in breast cancer. **a** OncoPrint of *RET* alterations in breast cancer. A case-by-case comparison of 121 breast cancers (column) carrying either a *RET* activating or uncharacterized rearrangement, missense mutation, or amplification (rows) to their *ERBB2* amplification status (row) by clinical genomic profiling and available ER (Estrogen Receptor), PR (Progesterone Receptor) status (rows) from routine immunohistochemistry, generated using the oncoprint software[58,59]. Only one case harbored both a *RET* rearrangement and a *RET* amplification. *RET protein expression in tissue from case with *RET* amplification is verified by western blot (Supplementary Fig. 4c). **b** *RET* fusions. Exon composition comparing full-length, wild-type *RET* with activating fusions identified in this study. Activating fusions maintain the exons (12–19) required for an intact kinase domain. Six patient samples carried the *CCDC6-RET* fusion. *NCOA4-RET* and a novel *RASGEF1A-RET* fusion initially detected in the index cases are functionally characterized in this report. UTR untranslated region, TM Transmembrane domain. **c** *RET* rearrangements. Exon composition of rearrangements including *RET* exons fused with novel partner genes (*ZNF485, RASGEF1A*), rearrangements of kinase domain-coding exons 12–19 of *RET* into intergenic space, rearrangements resulting in tandem duplications that involve exons 12–19 of *RET*. **d** *RET* point mutations. Schematic depicting location and number of the 25 *RET* mutations. Filled triangles and bold font indicate characterized activating mutations based on literature and open triangles represent uncharacterized mutations that have been described as somatic in cancer. Number in bracket represents number of cases. **e** Illustration of index case fusions (*NCOA4-RET* and *RASGEF1A-RET*) depicting breakpoints in exon 8 for *NCOA4* and intron 11 for *RET* resulting in a product encoding *NCOA4* (exons 2–7) fused to *RET* (exons 12–19). Breakpoints in intron 1 of *RASGEF1A* and intron 9 of *RET* modeled to result in an N-terminally truncated product ΔRET, as exon 1 of *RASGEF1A* is part of 5′ UTR and exon 11 of *RET* contains a potential alternate start site. CC coiled-coil domain, UTR untranslated region, ATG methionine start codon

mutations were analyzed to determine germline versus somatic variant status based on allele frequencies, altered copy number, and tumor purity and further verified using a second algorithm in cases where ploidy was two (Supplementary Table 2)[24,25]. Out of 25 missense mutations, 12 were germline and 7 were somatic. The remaining six were categorized as ambiguous and require further analysis using other approaches.

Two fusions were further analyzed in detail. The *NCOA4-RET* fusion was detected in an ER+ /PR−/HER2+ breast cancer and results from tandem duplication with breakpoints in *NCOA4* exon 8 and *RET* intron 11; includes the *NCOA4* exons encoding a putative coiled-coil domain and the *RET* exons encoding the kinase domain and therefore retains all the functional domains characteristic of activating NCOA4-RET fusion proteins that have

been studied in PTC and NSCLC[3,17,20,26] (Fig. 1e). The novel *RASGEF1A-RET* fusion was detected in an ER−/PR−/HER2− breast cancer and results from an inversion event on chromosome 10 with breakpoints in *RASGEF1A* intron 1 and *RET* intron 9 that juxtaposes the 5′UTR of *RASGEF1A* upstream of the *RET* kinase domain.

To characterize the novel *RASGEF1A-RET* fusion, exons 10–19 of *RET* were analyzed for the presence of an alternate internal start site. A methionine codon in exon 11 was identified that could potentially translate an N-terminally truncated RET protein with an intact RET kinase domain and this variant was designated as ΔRET. *RASGEF1A* does not contribute to the amino-acid sequence of ΔRET as it only encompasses the *RASGEF1A* 5′UTR. ΔRET may also serve to model some of the

**Table 1 Clinicopathologic characteristics of *RET* altered breast cancers**

| Characteristics | All breast cancer cases | *RET* altered cases | *RET* activating rearrangement | *RET* uncharacterized rearrangement | *RET* missense mutation | *RET* amplification |
|---|---|---|---|---|---|---|
| Number of cases (*n*) | 9693 | 121 | 8 | 8 | 25 | 81 |
| Median age (range), years | 54 (20–88) | 56 (31–85) | 61.5 (54–66) | 60 (48–69) | 52 (33–71) | 54 (28–85) |
| Median tumor mutational burden (range), mutations/Mb | 3.6 (0–251.4) | 4.5 (0–36.6) | 5.2 (0.9–7.2) | 4.5 (1.8–17.1) | 3.6 (0–36.6) | 4.5 (0–16.2) |
| Breast (*n*) as site of origin of sequenced sample | 3859 | 49 | 4 | 4 | 7 | 34 |
| Sequenced sample from a metastatic site (*n*) | 5834 | 72 | 4 | 4 | 40 | 47 |
| ER positive | NA | 34 (34.7%) | 2 (25%) | 2 (25%) | 15 (71.4%) | 16 (25%) |
| ER negative | NA | 64 (65.3%) | 5 (75%) | 5 (75%) | 6 (28.6%) | 48 (75%) |
| ER unknown | 8221 | 23 | 1 | 1 | 4 | 17 |
| *ERBB2* amplified | 1019 (10.5%) | 22 (18.2%) | 1 (12.5%) | 1 (12.5%) | 5 (20.0%) | 15 (18.5%) |
| *ERBB2* nonamplified | 8674 (89.5%) | 99 (81.8%) | 7 (87.5%) | 7 (87.5%) | 20 (80.0%) | 66 (81.5%) |

*Italics represent gene encoding RET, ERBB2*
*ER estrogen receptor status measured by routine clinical immunohistochemistry, ERBB2 gene amplification status by genomic profiling; ER and ERBB2 represented as numbers (%), NA not applicable, Mb megabase*

intergenic *RET* rearrangements (Fig. 1c) which are missing exons 1–10 but preserve kinase domain-coding exons 12–19. If using an internal methionine codon in exon 12, the intergenic rearrangements would also be N-terminal truncation mutants, similar to ΔRET, but missing more N-terminal residues from exon 11 and some kinase domain residues from exon 12. Since they are intergenic rearrangements, unlike *RASGEF1A-RET*, they could also potentially use alternate transcription initiation mapping prior to *RET* exons and, if in-frame, may include an intact RET kinase domain.

**Breast cancer RET fusions are constitutively active**. To assess the functionality of *NCOA4-RET* and *RASGEF1A-RET*, fusion constructs for *NCOA4-RET* and open reading frame (ORF) for ΔRET (product of *RASGEF1A-RET*) were developed and expressed in immortalized mouse fibroblasts NIH/3T3 and nontumorigenic MCF10A human epithelial mammary cells. Full-length, wild-type *RET* was ectopically expressed in order to model *RET* amplification leading to overexpression and is hereby referred to as RET$^{amp}$. Western blot analysis using an antibody against C-terminus of RET detected NCOA4-RET and ΔRET bands at predicted sizes of 68 and 46 kDa in the transduced cells respectively (Fig. 2a). To assess the oncogenic potential of the fusions, nontumorigenic NIH/3T3 and MCF10A cells expressing empty vector, RET$^{amp}$, ΔRET, or NCOA4-RET were evaluated for cell growth and colony formation. Expression of both fusions and RET$^{amp}$ resulted in a significantly increased growth capacity (Fig. 2b and Supplementary Fig. 1a). Increased clonogenic expansion was observed in NIH/3T3 cells when compared to vector control (Fig. 2c). In the absence of growth factor stimulation, expression of NCOA4-RET, ΔRET, and RET$^{amp}$ resulted in phosphorylation at tyrosine 905 (kinase activation loop tyrosine of RET) and tyrosine 1062 (major signaling hub of RET kinase to MAPK and PI3K-AKT pathways) consistent with constitutive kinase activation (Fig. 2d and Supplementary Fig. 1b). The kinase inactive mutant, K758M-RET, was used as a negative control and the constitutively active kinase mutant, M918T-RET, was used as a positive control. Increased signaling in the presence of RET alterations was confirmed for either RAS-MAPK pathway by phosphorylation of MEK or PI3K-AKT pathway by downstream phosphorylation of P70 S6 kinase (Fig. 2e).

**Breast cancer RET fusions confer sensitivity to RET inhibition**. RET fusions were evaluated for sensitivity to FDA-approved kinase inhibitors known to have activity against RET (Fig. 3 and Supplementary Fig. 2). Cabozantinib and sorafenib effectively reduced viability of NIH/3T3 and MCF10A breast cell lines expressing both RET fusions in an antiproliferative MTS assay (Fig. 3a and Supplementary Fig. 2a). In order to account for differences in growth rate, cell numbers were plated differentially for NIH/3T3 cells to ensure 80% confluence in vehicle-treated wells on the day of viability measurement. In the case of MCF10A, equal numbers were plated for all cell lines since growth differences were not observed between the cell lines at day 4 of the MCF10A growth assay, which coincides with the time duration from plating to reading of the MTS assay and does not confound the drug response results observed (Supplementary Fig. 1a). Further evaluation with cabozantinib in RET fusion cell lines by incubating with increasing concentrations of the drug revealed a dose-dependent reduction in phosphorylation of RET fusion kinase and downstream signals MEK and P70 S6 by western blot (Fig. 3b, Supplementary Fig. 2b) suggesting that the effect was driven by RET inhibition. Further, inhibition of downstream signaling was enhanced in NCOA4-RET cells in comparison to vector cells, which are negative for RET expression by western blot, thus suggesting the effect was mainly driven by targeting the RET fusion (Supplementary Fig. 3a). In addition, cabozantinib and another RET inhibitor, vandetanib, were able to effectively inhibit the enhanced colony-forming abilities of RET fusion cell lines and resulted in a significant reduction in size and number of colonies in a dose-dependent fashion in comparison to no-drug treatment and control cell lines (Fig. 3c, Supplementary Fig. 3b).

**RET fusions are tumorigenic**. NIH/3T3 cells transduced with RET fusions formed tumors in athymic nude mice within 2 weeks of subcutaneous injections (1 million cells, bilateral flank injections) for NCOA4-RET and 5 weeks for ΔRET, whereas vector and RET$^{amp}$ cells did not form tumors when examined for 10 weeks (Fig. 4a). To further explore tumor formation for RET$^{amp}$ cells, higher numbers (4.5 million cells, bilateral flank) were injected into NOD/SCID/interleukin 2 receptor γ null mice, which is a more powerful immunodeficient model compared to athymic nude mice. RET$^{amp}$ mice formed tumors within 7 weeks

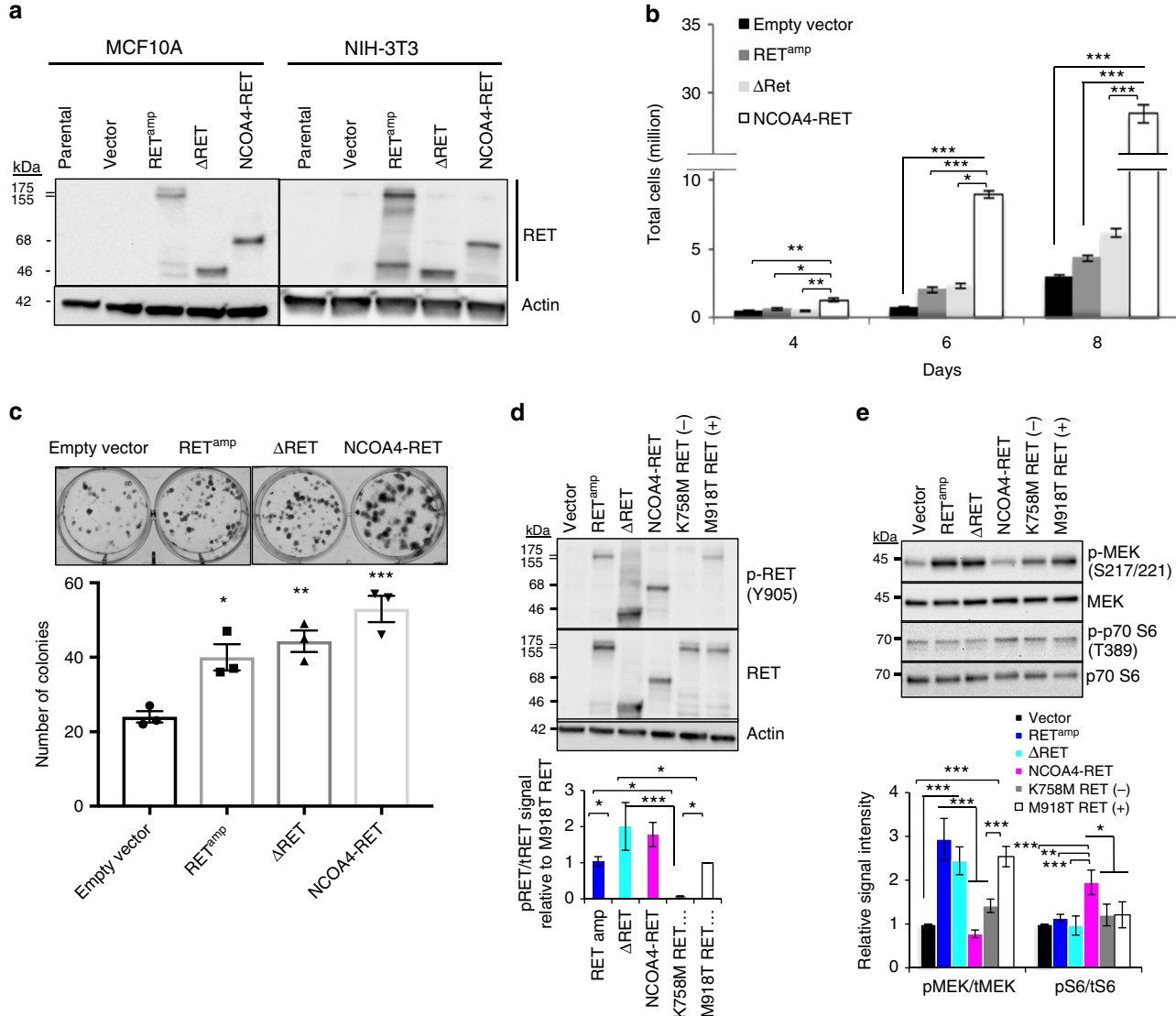

**Fig. 2** Transforming activity and oncogenic signaling of RET alterations. **a** MCF10A (human breast epithelial) and NIH/3T3 (immortalized mouse fibroblasts) cells overexpressing RET wild-type (RET[amp]), NCOA4-RET, and ΔRET. Expressed proteins were detected using a C-terminal RET antibody at predicted sizes of 155/175, 68, and 46 kDa respectively. NIH/3T3 cells transduced with RET[amp], NCOA4-RET, and ΔRET show increased **b** growth rates and **c** clonal expansion compared to cells transduced with vector alone. For growth curve experiments, 20,000 cells were plated per dish in triplicate for all cell lines and counted at days 4, 6, and 8. For clonogenic studies, 150 cells were plated per well in triplicate for all cell lines and stained with crystal violet at the end of 14 days. **d** Immunoblot analysis of NIH/3T3 cells overexpressing RET[amp], ΔRET, and NCOA4-RET reveal phosphorylation at tyrosine 905 and **e** downstream signaling measured after serum starvation for 24 h. In (**d**) and (**e**), kinase inactive mutant (K758M) and constitutively active mutant (M918T) refer to full-length RET variants used as negative and positive controls respectively. Results shown are representative of experiments performed thrice and error bars indicate s.d. ($n = 3$). $p \leq 0.05$ (*), $\leq 0.01$ (**), $\leq 0.001$ (***) are statistically significant and analyzed by ANOVA and Tukey's multiple comparisons test. Open-ended brackets depict comparison between the indicated group(s) and each of the groups under the bracket. Where brackets are absent, comparison is with empty vector

whereas matched vector cells did not form tumors when examined for 10 weeks (Fig. 4b). Together, the data in Fig. 4a, b suggest that while RET fusions are tumorigenic, constitutive activation of the RET kinase by RET overexpression alone (RET[amp]) is also capable of driving tumor growth under sufficiently permissive conditions. Immunohistochemistry revealed positive staining for the proliferation marker Ki-67 for RET[amp], ΔRET, and NCOA4-RET tumor tissues (Fig. 4c). Tumor protein was verified for the expression of RET fusions by immunoblot and activation of PI3K-AKT and MAPK pathways was observed (Fig. 4d). Based on experiments described in NIH/3T3 cells (Figs. 2b and 4a), NCOA4-RET was observed to be the fastest growing cell line and

this translated to tumor formation in less than 2 weeks; whereas ΔRET and RET[amp] were slower in comparison, taking longer than a month. Next, the effect of RET inhibitor, cabozantinib on tumor response was tested in each xenograft model. In order to offset the time for tumor formation, $0.5 \times 10^6$ cells were injected for NCOA4-RET, whereas $5 \times 10^6$ cells were injected for ΔRET and RET[amp] in athymic nude mice.

**RET-fusion-driven tumors respond to a RET inhibitor**. Cabozantinib was used in xenograft models for NCOA4-RET, ΔRET, and RET[amp] to determine drug sensitivity in vivo. In each case,

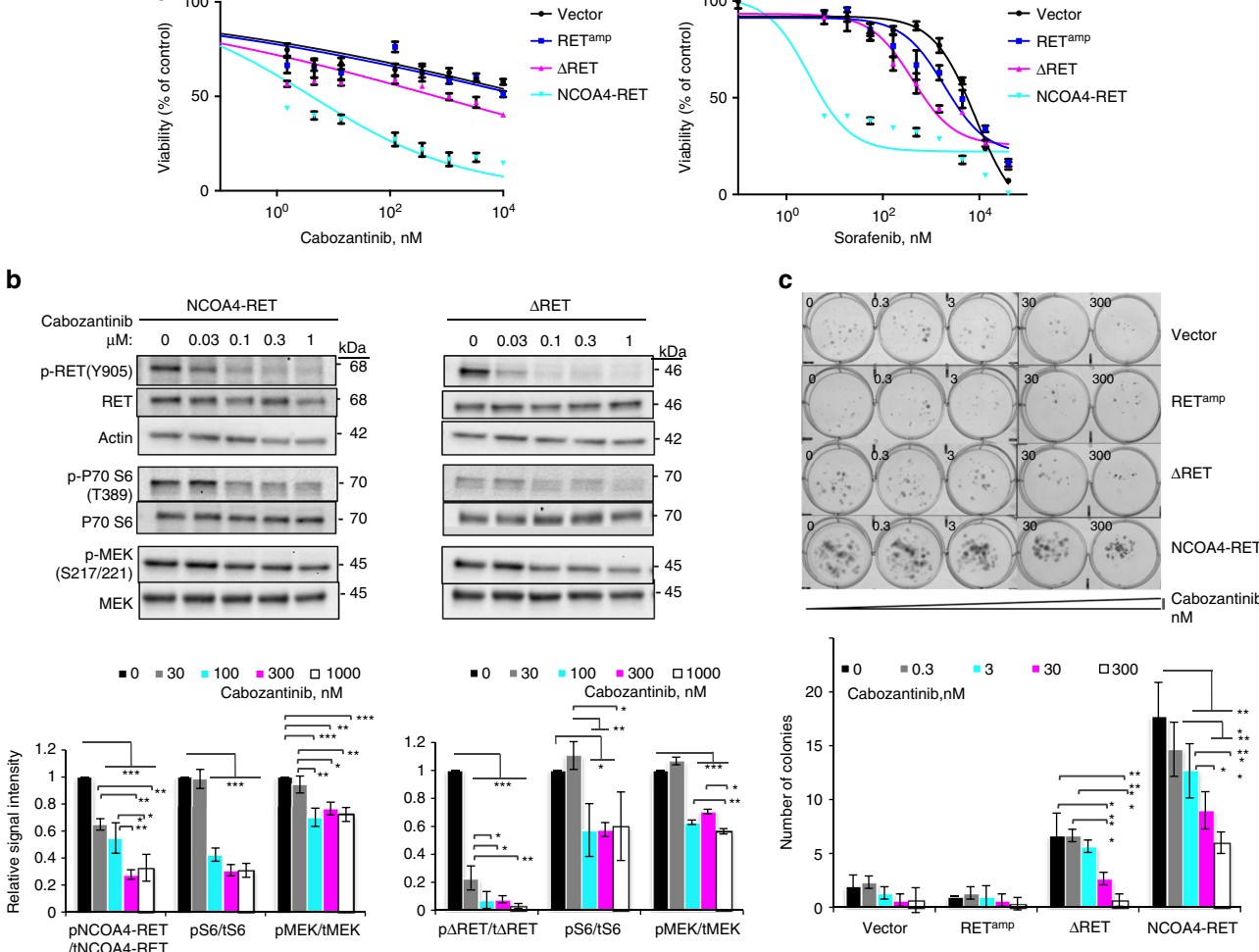

**Fig. 3** Fusion cell lines exhibit dose-dependent response to RET inhibition. **a** Dose−response curves after 72 h of drug treatment with cabozantinib or sorafenib in NIH/3T3 cells expressing RET[amp], ΔRET, NCOA4-RET, and vector. Cell viability normalized to vehicle (DMSO)-treated cells. Error bars indicate s.d. of three replicates and are representative of three independent experiments ($n = 3$). **b** Western blot indicating inhibition of RET fusion kinase, MEK and P70 S6 signaling with increasing concentration of cabozantinib in NIH/3T3 cells transiently expressing NCOA4-RET or ΔRET. Measurements were made after overnight serum starvation and 1 h of incubation with cabozantinib in the absence of serum. Graphs represent image densitometry analysis of western blots from three independent experiments ($n = 3$). Ratio of phosphorylated to total proteins is measured at each concentration and mean values with error bars indicating s.d. are plotted relative to DMSO-treated control. **c** Dose-dependent reduction in ΔRET and NCOA4-RET colony numbers upon treatment with increasing concentrations of cabozantinib in NIH/3T3 transduced cells for 14 days. 0 represents DMSO-treated controls. All cells were plated at equal numbers per well in triplicates per experiment. Error bars indicate s.d. of three replicate measurements per condition ($n = 3$) and are representative of experiments performed three times. $p \leq 0.05$ (*), $\leq 0.01$ (**) and $\leq 0.001$ (***) by ANOVA (one-way for (**b**) and two-way for (**c**)) with Tukey's multiple comparisons test. Open-ended brackets depict comparison between the indicated group and each of the groups under the bracket. Where brackets are absent, comparison is with DMSO control

upon tumor formation, mice were randomized into three groups and treated with either a low dose (30 mg kg$^{-1}$), high dose (60 mg kg$^{-1}$) of cabozantinib, or vehicle saline. Treatment with 60 and 30 mg kg$^{-1}$ effectively inhibited tumor growth for NCOA4-RET xenografts and resulted in rapid regression of the tumor within 2 weeks (Fig. 5a). In case of ΔRET, both doses inhibited tumor growth and a dose-dependent effect was observed, with the higher dose of 60 mg kg$^{-1}$ resulting in a more significant reduction in tumor volume when compared to 30 mg kg$^{-1}$ at the end of 2 weeks (Fig. 5b). RET[amp] xenografts also showed significant reduction in tumor volumes with both doses of cabozantinib in comparison to vehicle, despite increased variability between mice from the same treatment group (Supplementary Fig. 4a and 4b). Overall, results indicated that all three RET alterations are sensitive to cabozantinib as evidenced by significant tumor volume reduction.

Tumor protein, collected at the end of treatment on day 14 for NCOA4-RET and ΔRET, revealed significant reduction in the fusion protein levels and downstream PI3K-AKT signaling for the cabozantinib-treated groups as measured by western blot (Fig. 5c and Supplementary Fig. 4d). Histological analysis of the xenograft tissue for NCOA4-RET and ΔRET shows a stark contrast in tumor cellularity between vehicle and treatment groups (Fig. 5d), supporting the reduction of fusion proteins observed in Fig. 5c. A packed population of highly proliferative cells as revealed by Ki-67 stain in vehicle tissue is efficiently cleared in the treatment groups in a dose-dependent manner. The treated tumors reveal hyalinization and apoptosis. Treated tumor tissues stained positive for Cleaved Caspase-3 whereas vehicle tissue was relatively negative verifying activation of the apoptotic pathway for tumors treated with cabozantinib (Fig. 5d). Xenograft data with RET[amp] are suggestive of a benefit; however, the large

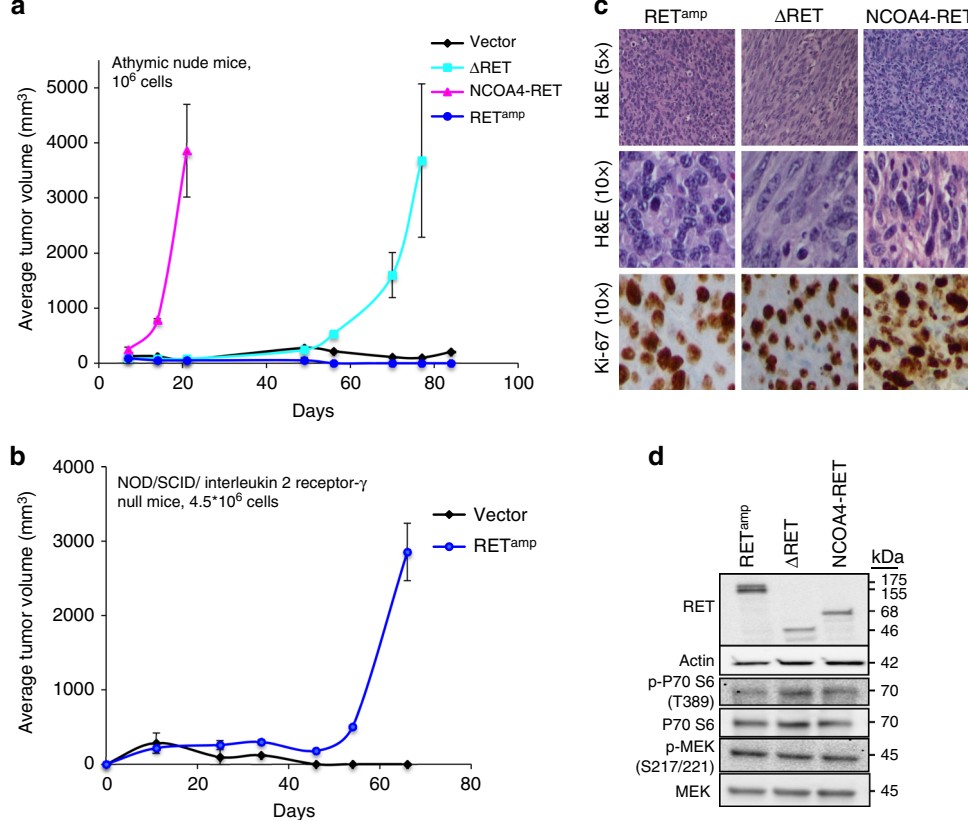

**Fig. 4** ΔRET and NCOA4-RET are tumorigenic. Growth curve of tumors formed upon subcutaneous injection of transduced NIH/3T3 cells at **a** $1\times10^6$ cells, bilateral flank injections in athymic, nude mice ($n = 4$ injection sites for vector, ΔRET, and RET[amp] and $n = 6$ injection sites for NCOA4-RET) and **b** >$4.5\times10^6$ cells, bilateral flank injections in NOD/SCID/ interleukin 2 receptor γ null mice ($n = 4$ injection sites per group) for vector and RET[amp] cells. Error bars represent mean ± s.d. **c** Representative staining for hematoxylin and eosin (H&E) demonstrates a packed population of tumor cells (top row, ×5 magnification). At higher power (middle row, ×10 magnification) tumor cells reveal mitoses. Tumor cells stain positive after immunohistochemistry for proliferation marker Ki-67 (bottom row, ×10 magnification). **d** Immunoblot of tumor protein lysates from NIH/3T3 xenografts using the C-terminal RET antibody and downstream signaling proteins

confidence intervals and smaller differences compared to control should be interpreted more cautiously and warrants further exploration (Supplementary Fig. 4a and 4b).

**NCOA4-RET positive breast cancer responds to cabozantinib.** A 63-year-old female with a remote history of a stage I, ER− breast cancer (24 years prior) and a more recent stage I, ER +/HER2− breast cancer (13 years prior, Fig. 6a) was found to have stage IV disease in 2014. The patient initially had a lumpectomy followed by adjuvant radiation only, but required a mastectomy with axillary lymph node dissection for the second occurrence which was in the same breast followed by four cycles of doxorubicin and cyclophosphamide and then by 5 years of tamoxifen. In 2014, a regional/distant recurrence of ER+/HER2+ (by immunohistochemistry) disease (Fig. 6b) was confirmed on ultrasound-guided biopsy of the right axillary tail and both MRI and PET/CT imaging showing bony metastasis. Biopsy tissue was sent for genomic profiling (FoundationOne, Foundation Medicine) that identified the NCOA4-RET fusion and, based on the breakpoints, was found to join exons 1−7 of NCOA4 in frame with exons 12−19 of RET. Histology of the recurrent breast cancer tissue revealed a papillary architecture which is characteristic of RET fusion-positive thyroid cancers. Palliative radiation to the thoracic-lumbar spine followed by HER2-targeted treatment with pertuzumab, trastuzumab, and

anastrazole was initiated. Subsequently, axillary and bony progression was found and second-line therapy with combined trastuzumab, exemestane, and cabozantinib was initiated. Cabozantinib was added as a genomic-guided targeted therapy based on the NCOA4-RET fusion observed in the recurrent tumor. In preclinical studies, NCOA4-RET fusion-driven models have previously been targeted with RET inhibitors[27] and clinical studies have demonstrated responses to RET inhibitors for patients with RET fusions in NSCLC[28–30], which further supported treatment for this patient. The patient was treated initially with 140 mg day$^{-1}$ of cabozantinib but due to side effects such as shortness of breath, intermittent treatment and dose reduction to 100 mg day$^{-1}$ was required, and ultimately, treatment had to be discontinued (Fig. 6c). For a total of 67 days on cabozantinib, which amounts to 80% of duration for second-line treatment, there was a rapid radiographic and clinical response along with improvement in dyspnea and epistaxis. Representative PET/CT images of the thoracic spine lesion (Fig. 6d), which was present prior to cabozantinib, showed a significant reduction in PET avidity after treatment. Whether the response was due solely to cabozantinib, a change in hormone treatment, or the combination of these therapies cannot be determined with certainty. However, addition of a RET inhibitor to treatment of an ER+/HER2 treatment-refractory, NCOA4-RET- positive breast cancer resulted in a clinical response.

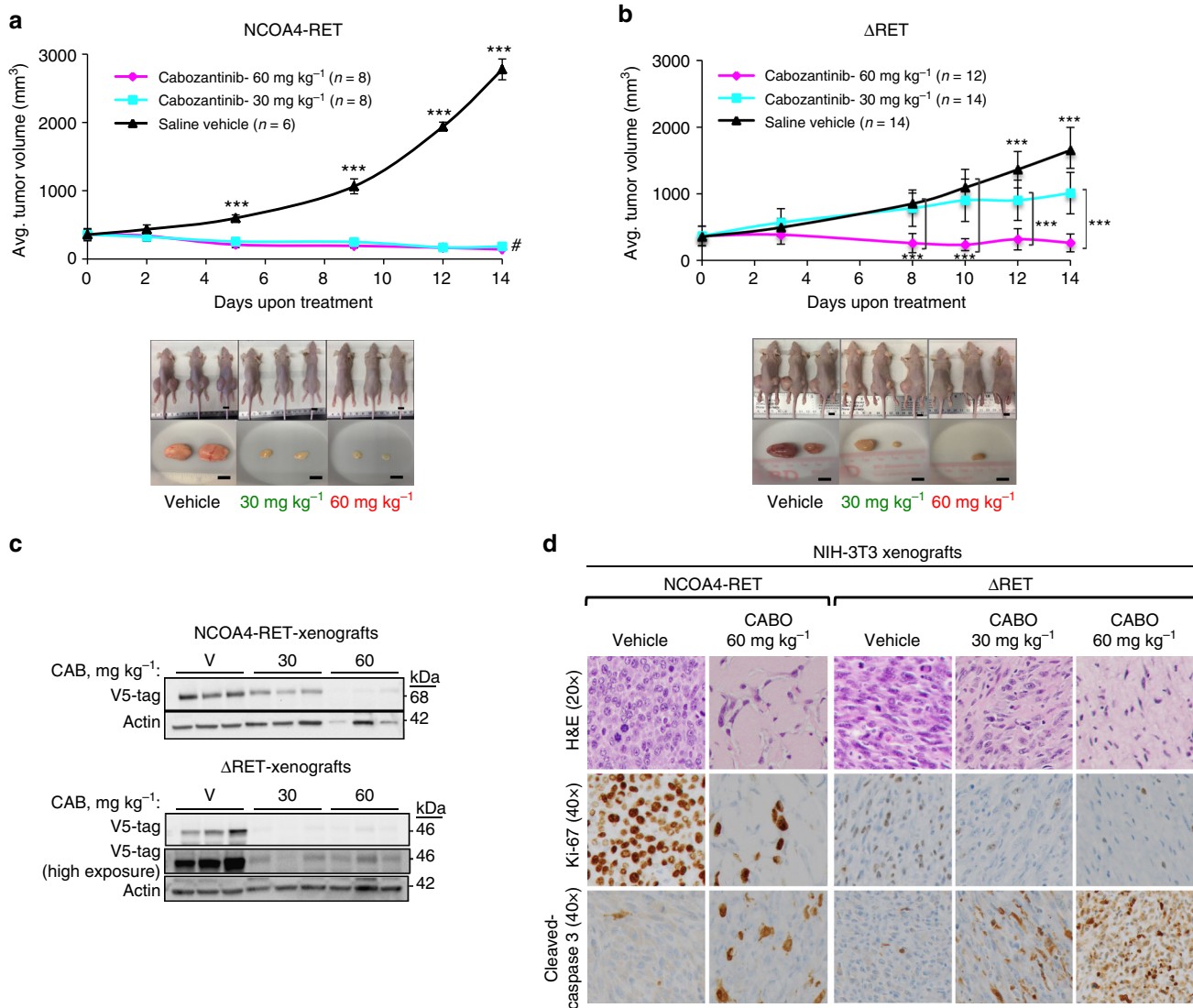

**Fig. 5** Cabozantinib inhibits tumor growth driven by RET fusions. Mean tumor volume was measured in NIH/3T3 xenograft tumors driven by **a** NCOA4-RET ($n = 8$ or 6 injection sites per group) or **b** ΔRET ($n = 14$ or 12 injection sites per group) under treatment with either cabozantinib at 30 mg kg$^{-1}$, 60 mg kg$^{-1}$, or saline vehicle control for 14 days. Treatment started at day 0. Error bars represent mean ± s.d., ***($p \leq 0.001$) represent comparisons between both treatment groups and vehicle-treated controls when depicted above the vehicle curve or between indicated groups and #($p \leq 0.001$) represents comparisons between day 0 and day 14 for the treatment groups (two-way ANOVA with Tukey's multiple comparison test). Mouse and tumor images are representatives from each treatment group at the end of study after 14 days of treatment. Scale bars indicate 10 mm. **c** Immunoblot of tumor protein lysates collected at the end of 14 days of treatment to measure changes in NCOA4-RET, ΔRET (detected by V-5 tag antibody, actin as loading control). Mice were treated on the day of tumor harvest for 4 h with saline vehicle or cabozantinib (30 or 60 mg kg$^{-1}$) and sacrificed. $n = 3$ xenograft samples per treatment condition. **d** Representative hematoxylin and eosin (H&E) staining of tumor tissue revealing a high grade sarcomatoid distribution (top row, ×20 magnification) and immunohistochemistry of markers Ki-67, Cleaved Caspase-3 for comparison between vehicle and cabozantinib treatment groups in NCOA4-RET and ΔRET xenograft experiments (middle and bottom row, ×40 magnification)

## Discussion

In this study, *RET* alterations are reported from a large cohort of 9693 breast cancers that were genomically profiled as part of routine clinical care. Analysis of the *RET* genomic landscape has not been previously reported at this scale exclusively for breast cancer. Large sample size, high sequencing depth (>600×), and hybrid capture to identify select introns in *RET* ensured accurate detection of all classes of genomic variants, robust statistical analyses as well as validation of rare variants resulting in a hit rate of 1.2%. While this may be a relatively small fraction of patients,

given that roughly 250,000 new cases of invasive breast cancer are diagnosed in the U.S. annually[31], this extrapolates to roughly 3000 new cases each year in which *RET* alterations may become potentially relevant therapeutic targets. This is similar to the frequency of *ALK* fusion-positive NSCLC diagnosed in the U.S. annually[32]. The fraction of advanced disease in this cohort is likely underestimated (Table 1) as these assays are often performed on breast tumor tissue when distant site tumor tissue is insufficient in quality or quantity. Considering the advanced or refractory nature of cases that undergo genomic profiling,

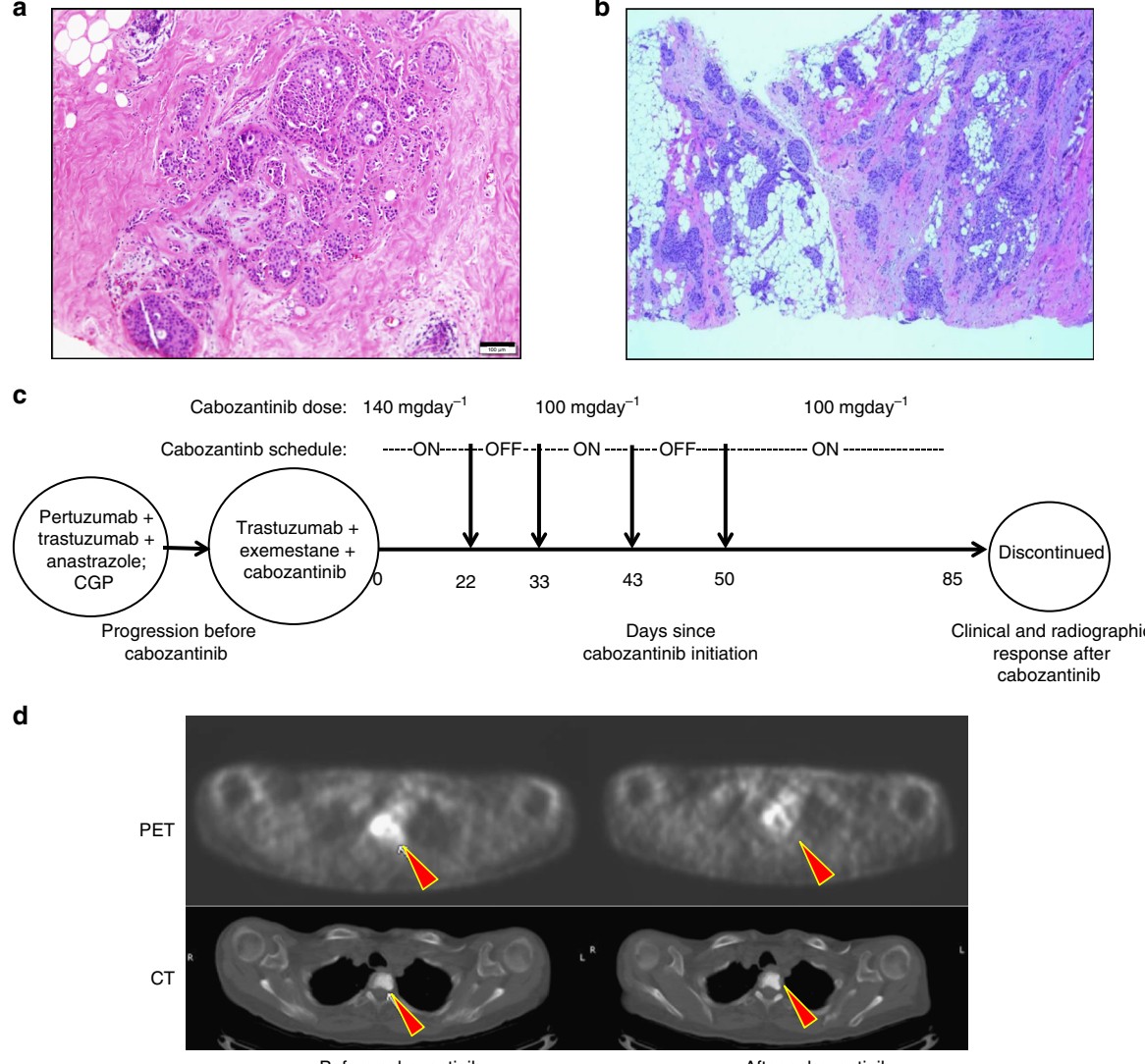

**Fig. 6** Clinical response to targeted therapy in a patient with RET-fusion+ve breast cancer. **a** Histology image of stage I ER+/HER2− tumor. Scale bar indicates 100 μm. **b** Histology of the regional/distant recurrence of ER+/PR−/HER2 3+ tumor from the right axillary tail, ×4 magnification. Genomic profiling on this tissue revealed *NCOA4-RET* fusion (same as patient 3 in Fig. 1b). **c** Treatment schematic and timeline showing initiation of cabozantinib (day 0) after progressing on HER2-targeted treatment (pertuzumab, trastuzumab) and anastrazole. Targeted genomic profiling identified the presence of *NCOA4-RET* fusion. Based on the finding, cabozantinib, a RET inhibitor, was added along with trastuzumab and exemestane as second-line treatment. Intermittent treatment and dose reduction was required due to side effects. Clinical and radiographic response was observed 85 days after cabozantinib initiation. **d** Representative PET (top row) and CT (bottom row) images of thoracic spine lesion (filled red arrows) before and after cabozantinib treatment. PET signal avidity is reduced after treatment

identifying a targetable alteration holds promise for these patients who may face diminishing options using standard approaches.

Resistance to cancer treatment can arise from tumor evolution resulting in development of fusions, mutations, or amplifications and alternative, by-pass signaling pathways driven by other tyrosine kinases including other receptor tyrosine kinases (RTK)[33,34]. For the *NCOA4-RET* fusion-positive, ER+/*ERBB2*-amplified breast cancer, one can speculate that the presence of *NCOA4-RET* contributed to its relative resistance to primary treatment with aromatase inhibitor, trastuzumab, and pertuzumab. While this patient had clinical benefit after switching to cabozantinib, trastuzumab, and an alternative aromatase inhibitor, one cannot determine with certainty, based on case information alone, whether that clinical benefit was primarily attributable to initiation of cabozantinib, the new aromatase

inhibitor, or the combination of these drugs. Intriguingly, co-existence of *NCOA4-RET* with another HER family member, EGFR (HER1), has been shown as a mechanism of resistance in an *EGFR*-mutated lung cancer treated with afatinib[35]. Of note is that at least one other *ERBB2*-amplified breast cancer in this cohort was known to have developed *RET* amplification after it had acquired resistance to multiple HER2-targeted agents and that this was associated with overexpression of RET in the tumor (* in Fig. 1a and Supplementary Fig. 4c). Modeling of amplification by overexpression of wild-type RET sequence validates that RET overexpression, similar to RET fusions, can lead to aberrant downstream signaling like other RTKs such as *ERBB2* amplification in breast cancer or *MET* amplification in lung cancer. The response of the RET$^{amp}$ xenografts to cabozantinib also reveals the potential for RET inhibition in treating *RET* amplification-

positive breast cancers although this needs to be corroborated with appropriate patient-derived xenograft models.

While interchromosomal rearrangements leading to *RET* fusions have been reported previously, all *RET* fusions detected in this cohort reflect intrachromosomal rearrangements in chromosome 10. The exclusive presence of chromosome 10 partners here may reflect loss of specific DNA damage and repair mechanisms particular to basal-like breast cancers, that have been known to exhibit numerous intrachromosomal rearrangements when compared to luminal subtypes[36]. However, the overriding challenge for many *RET* rearrangements is a need to model and characterize novel structural rearrangements to determine functionality before their identification can be incorporated appropriately for tailoring therapeutic approaches. Importantly, the *RASGEF1A-RET* fusion serves to model a number of novel rearrangements identified in this cohort. Other rearrangements warrant further evaluation. For instance, for four RET kinase domain duplications detected in this study (Fig. 1c), predicting ORFs with duplicated exons alone does not detect potential alternative splicing events that could lead to kinase domain duplications in the protein as algorithms recognize the native STOP codons. The recurrent finding of such duplications warrants detailed enquiry including intronic sequences and scoring of potential splice regulator sites using validated splice prediction tools. However, that there is a precedent for functional and therapeutically actionable EGFR, MET, and BRAF kinase domain duplications in other cancers is suggestive that a similar mechanism may be applicable to the observed RET kinase duplications in breast cancer[37–39].

Comparison of *NCOA4-RET* and *RASGEF1A-RET* in tumorigenic and in vitro assays reveal that both fusions induce transformation of nontumorigenic cells, but *NCOA4-RET* potentially confers a more aggressive oncogenic phenotype than *RASGEF1A-RET*. Similar to many *RET* missense mutations, where the oncogenic potential may differ from variant to variant[40], preclinical modeling of *RET* rearrangements shows differing severity of phenotypes associated with expression of the canonical CCDC6-RET and NCOA4-RET fusions[41]. Additionally, RET fusions that are under control of dimerizing partners (NCOA4, CCDC6) display distinctions in functional pathways employed[41]. This may further differentiate truncated kinases without fusion partners such as ΔRET from NCOA4-RET. The preference of one signaling pathway over another has been reported for NCOA4-RET (previously referred to as RET/PTC3 in thyroid cancer) where enhanced signaling occurs via AKT rather than ERK, although both pathways are activated[42]. This is also observed in the model in Fig. 2e. ΔRET is modeled after RASGEF1A-RET and differential tissue expression between RET and RASGEF1A can further drive oncogenic potential by influencing expression levels of ΔRET, which will be under the more ubiquitous RASGEF1A promoter in the breast tissue for this fusion. Moreover, rearrangements with less oncogenic potential, such as ΔRET, may become more potent in the setting of specific co-events similar to synergistic potentiation by co-events observed for B-cell lymphomas[43]. With increasing efficiency and accuracy of genome-editing tools such as CRISPR, it is anticipated that it would become easier to assess complex genomic rearrangements at more endogenous levels of expression and in the context of tissue type.

Characterization of the novel *RASGEF1A-RET* (ΔRET) fusion revealed that an N-terminally truncated RET kinase domain is oncogenic and targetable. Precedence for activated and targetable, N-terminally truncated tyrosine kinases has been reported for HER2 [44–46], ALK[47], c-KIT[48], and EGFR[49] which lack the extracellular and transmembrane regions in breast, melanoma, prostate cancers, and gliomas, respectively. ΔRET also serves to model other intergenic rearrangements of *RET* such as those identified in this cohort that may place kinase domain-coding exons under

the influence of alternate start sites, promoters, and enhancers, and could potentially render a larger group of breast cancers therapeutically actionable. The N-terminally truncated ΔRET, which includes residues 674−1072 of RET, includes a portion of the juxtamembrane region, an intact kinase domain and the C-terminal tails. The juxtamembrane (JM) domain is functionally important in regulating auto-inhibitory functions of RTKs[50,51]. Loss or changes in the JM region of RET have been associated with cancer. In-frame insertion/deletion mutations and single base changes in JM domain (666 and 691) have been reported to cause gain-of-function, constitutively active, monomeric RET in familial and sporadic medullary thyroid cancer.[21,52,53] With the loss of extracellular ligand-binding domain, transmembrane domain, and partial loss of the auto-inhibitory juxtamembrane domain, ΔRET is hypothesized to be active by auto-phosphorylation in a cytosolic and ligand-independent manner.

A more comprehensive understanding of the effect of *RET* amplification on tumorigenesis and drug sensitivity as well as whether this alteration translates into increased mRNA and protein levels is needed, given its overall frequency in breast cancer. A recent study has demonstrated that chronic over-expression of RET wild-type sequence in an inducible, transgenic mouse model results in luminal mammary tumors that are also responsive to a RET kinase inhibitor[54]. The minimally amplified region around *RET* in this breast cancer cohort cannot be determined from targeted genomic profiling alone. Therefore, the presence of an amplicon that encompasses another oncogene surrounding *RET* on chromosome 10 cannot be ruled out. Notably, while *RET* is the only compelling oncogene from the Cancer Gene Census on the 10q11 segment, additional evaluation of this region, similar to what has been done for 8p11 in breast cancer, is required[55]. This study highlights that the frequent finding of *RET* amplification, its near exclusivity with respect to other *RET* alterations reported here, response to TKI, and the known involvement of RET overexpression in tumorigenesis and resistance to hormonal therapies is worthy of follow-up in future clinical datasets and currently available databases with access to detailed genomic, treatment, and drug response histories.

Subsets of genomic alterations in RET were enriched in breast cancer subtypes. Of all variant classes, *RET* amplifications were the most commonly observed and mainly found in ER− and *ERBB2*− (*ERBB2* wild-type/HER2-) breast cancers, followed by missense mutations and rearrangements. There was a trend toward more *RET* rearrangements being found in ER− breast cancers. Conversely, *RET* missense mutations were more frequently associated with ER+ breast cancers. *RET* missense mutations are found as germline mutations in cancer susceptibility syndromes such as MEN2, but may also be somatic[56]. Algorithms using allele frequency, tumor purity, and ploidy may suggest whether or not a missense mutation is likely germline or somatic; however, clinical care guidelines would include genetic counseling and testing based on the strength of the patient's personal medical and family history[57]. As genomic profiling cases may be biased toward more aggressive or refractory breast cancers and this cohort was restricted to cases with available ER status, the distribution of *RET* alterations may be skewed and limited in generalization to specific breast cancer subtypes. Yet, analysis of publically available breast cancer datasets through cBioPortal reveals similar trends (Supplementary Fig. 5)[58,59]. The presence of *RET* missense mutations in ER+ breast cancers, a subset which may be hormone therapy resistant, becomes highly relevant and merits further study given known associations between RET expression levels and reduced sensitivity to hormonal therapies[9–11]. Further studies demonstrating any differential response to specific inhibitors based on type of RET alteration may help guide use of the most appropriate inhibitor.

Multi-kinase inhibitors with activity against RET such as sorafenib, vandetanib, and others have been tested in clinical trials in unselected breast cancers with modest to no significant activity[60,61]. However, both the presence of the target, e.g. RET alteration, and an understanding of the effect that any individual genomic alteration has on inhibitor response is critical. For instance, RET V804M is a known gatekeeper mutation that contributes to cabozantinib resistance, but tumors with this mutation may still retain sensitivity to other tyrosine kinase inhibitors with activity against RET such as ponatinib and AD80 based on preclinical data[62,63]. Moreover, preclinical modeling of CCDC6-RET and NCOA4-RET demonstrated distinct TKI sensitivities[41]. Recent clinical trials have shown that RET inhibitors are active in a subset of patients with lung cancers harboring RET-rearrangements[5,29]. Novel approaches to target RET rearrangements are now being evaluated by developing potent inhibitors with high selectivity and improved biochemical profiles[27,63]. With new evidence of the presence of oncogenic, recurrent RET fusions and rearrangements in breast cancer, clinical trials catering to RET-rearranged cancers need to be expanded to evaluate these selected breast cancers.

In summary, recurrent canonical and noncanonical RET alterations are observed in breast cancer. Supporting clinical considerations for targeting RET rearrangements are the functional activities attributable to the NCOA4-RET fusion and to the novel RASGEF1A-RET fusion, which are the focus of this report. Results support the oncogenic potential of both types of RET fusions as well as that of RET amplification. That RET alterations are sensitive to RET inhibitors, as demonstrated in vitro and in vivo models as well as in a patient, suggest that they are potentially actionable in the subset of breast cancers in which they are found. Further studies are warranted to better understand the frequency and correlation of RET alterations with response to standard therapies, and the ideal strategies to therapeutically target RET and improve patient outcomes.

## Methods

**Study population**. Formalin-fixed, paraffin-embedded (FFPE) tissue samples obtained from a total available $n = 9693$ patients with breast cancer were submitted by clinicians for targeted next-generation sequencing as part of routine clinical care. The pathologic diagnosis of each submitted case was confirmed and tumor content was determined on routine hematoxylin- and eosin-stained slides. Specimens were submitted with limited accompanying clinical information; pathology reports were reviewed to determine ER status where available. Approval for this study, including a waiver of informed consent and an HIPAA (Health Insurance Portability and Accountability Act of 1996) waiver of authorization, was obtained from the Western Institutional Review Board (Protocol No. 20152817). The study complied with all ethical regulations relevant to human research participants and data.

**DNA and RNA sequencing**. Targeted next-generation sequencing was performed in a Clinical Laboratory Improvement Amendments (CLIA)-certified, New York State-accredited, and College of American Pathologists (CAP)-accredited laboratory (Foundation Medicine, Inc., Cambridge, MA) between May 2012 and Sept. 2016. Technical details and validation of genomic profiling assays have been extensively described[64,65]. In brief, at least 50 ng DNA was extracted from 40-μm FFPE sections with at least 20% tumor cells. Adaptor-ligated DNA underwent hybrid capture for all coding exons of 236, 315, or 405 cancer-related genes plus select introns from 19, 28, or 31 genes frequently rearranged in cancer (Supplementary Data 1 for gene list). Captured libraries were sequenced to a median exon coverage depth of 637× (Illumina HiSeq). Resultant sequences were analyzed for short variants (base substitutions, insertions/deletions (indels)), copy number alterations (focal amplifications and homozygous deletions), and select gene fusions/rearrangements. RNA sequences were analyzed for the presence of fusions/rearrangements only. Custom filtering was applied to report genomic alterations and remove benign germline events[64,65]. Copy number detection and validation was obtained by normalizing the reads in the tumor against a process-matched normal control and further corrected for tumor purity[64]. Germline versus somatic variant status for missense mutations were determined based on allele frequencies, altered copy numbers, and tumor purity[24,25].

**Fusion sequences and expression plasmids**. The human fusion ORF cDNA sequences for NCOA4 were amplified from MCF7 (ATCC® HTB-22™) cell line and ΔRET was amplified from pDONR223-RET, a gift from William Hahn & David Root (Addgene plasmid #23906) using recombinant Taq DNA polymerase enzyme (Life Technologies) with fusion-specific primers (Supplementary Table 3). Gel-extracted PCR products for NCOA4-RET were ligated using T4 DNA ligase enzyme (Invitrogen). The predicted ORFs for NCOA4-RET, ΔRET and full-length, wild-type RET (RET$^{amp}$) were cloned into pLenti6.3/V5–DEST and pEF-DEST51 vectors (Invitrogen) that include a C-terminal V5-tag. Constitutively kinase active (corresponding to M918T) and kinase inactive (corresponding to K758M) full-length variants were generated using QuikChange II XL Site-Directed Mutagenesis Kit (Agilent). All constructs were verified by DNA sequencing.

**Cell lines and in vitro overexpression of ORFs**. Human breast cell line MCF 10A (ATCC® CRL-10317™) and mouse fibroblast cell line NIH/3T3 (ATCC® CRL-1658™) were obtained from American Type Culture Collection (ATCC) and cultured according to ATCC recommended protocols. Authentication of all cell lines was performed by ATCC using short tandem repeat analysis and mycoplasma contamination tested negative. Both MCF10A and NIH/3T3 cells lacked mRNA expression of RET co-receptor, GFRα as verified by qPCR and detectable RET protein as measured by western blot. The pLenti6.3 vectors were used for over-expression in MCF10a using Fugene HD transfection (Promega) and in NIH/3T3 cells using Lipofectamine 3000 (Invitrogen). pEF-DEST51 vectors were transduced into NIH/3T3 and selected with blasticidin (Invitrogen). For all cell line experiments and growth curves, cells were counted using Vi-CELL XR (Beckman Coulter) cell analyzer using trypan blue die exclusion method. For measurements involving phosphorylated proteins, NIH/3T3 and MCF10A cells were serum starved for 24 h in DMEM media before protein isolation, which is described in detail below. For all other experiments, NIH/3T3 cells were cultured with 10% bovine calf serum in media and MCF10A cells were cultured in ATCC-recommended mammary epithelial growth media that includes mammary epithelial basal media supplemented with bovine pituitary extract, human epidermal growth factor, insulin and hydrocortisone (MEGM-kit, Lonza, CC-3150).

**Western blot analysis**. Protein samples were extracted from cells and tumor xenografts using NETN lysis buffer (Tris 20 mM pH 8.0, NaCl 150 mM pH 7.5, EDTA 1 mM, NP40 0.5%) along with protease and phosphatase inhibitor cocktails (Sigma-Aldrich). For lysates from xenografts tumors, frozen tissue was weighed and approximately 20 mg was homogenized in 200 μl lysis buffer on ice. After homogenization, samples were briefly sonicated twice for 20 s each and incubated on ice for 30 min, followed by centrifugation at 14,000 rpm for 15 min at 4 °C. For lysates from cells, samples were vortexed for 5 s every 5 min while incubating on ice for 15 min. This was followed by centrifugation at 14,000 rpm for 10 min at 4 °C. All lysates were separated by gradient (4–12%) sodium dodecyl sulphate-polyacrylamide gel electrophoresis (SDS-PAGE) under reducing conditions and transferred onto 0.2 μm polyvinylidene fluoride membrane for detection. The dilutions of primary antibodies used were 1:250−1:1000 for anti-phospho-Ret (Y905) (Cell Signaling Technology, 3221), 1:250 for anti-phospho-Ret (Y1062) (Santa Cruz Biotechnology, 20252-R) and 1:1000 for anti-Ret specific to the C-terminal region (Cell Signaling Technology, 14698). For all other rabbit primary antibodies (phospho and total MEK (9154, 9126), P70 S6 (9234, 2708) and total V5-tag (13202)), dilutions were used at 1:1000 (Cell Signaling Technology). Mouse anti-actin was used at 1:10,000 (Sigma-Aldrich, clone AC-15). The full list of antibodies and details are included in Supplementary Table 4. Enhanced chemi-luminescence (ECL) reagent was added to the membrane (Pierce Biotechnology) and bands were detected on the ChemiDoc™ Imaging System. Image densitometry was performed using the Image Lab software (Bio-Rad). For analyzing fusion-driven signaling, cells were serum-starved for 24 h prior to protein extraction. To study changes in fusion signaling under the influence of RET inhibitor cabozantinib (Selleckchem), cells were serum-starved overnight and treated with cabozantinib for 1 h and compared to vehicle DMSO-treated controls. Uncropped scans of western blot images in Figs. 2 and 3 are shown in Supplementary Figs. 6-7.

**Cell viability assay**. Cell viability was measured using the CellTiter Aqueous MTS reagent (Promega) according to the manufacturer's protocol. Following optimization and accounting for growth rate differences, 1000 NCOA4-RET cells and 2000 vector, RET$^{amp}$ and ΔRET cells were allowed to attach per well in 96-well plates overnight for NIH/3T3 cells. For MCF10A, all cells were plated at 5000 per well. Cells were then treated with increasing concentration of inhibitors, cabozantinib or sorafenib (Selleckchem) for 72 h. After incubating with MTS reagent for 2–4 h at 37 °C, absorbance was measured at 490 nm on a plate reader (Infinite 200 PRO, Tecan). Viability of cells was calculated by dividing the average absorbance of treatment wells by that of vehicle DMSO-treated wells. Data were analyzed and represented using GraphPad Prism version 7 software.

**Clonogenic assay**. Cells were seeded at 150 per well in six-well plates and incubated for 14 days. Colonies were stained with crystal violet (2% solution, Sigma-Aldrich) and visualized using ChemiDoc™ Imaging System (Bio-Rad). Colonies greater than 50 cells were counted using a cell counter (ImageJ) for comparison

between groups. For experiments involving inhibitor treatment, cells were incubated with inhibitor (cabozantinib, vandetanib (Selleckchem)) or DMSO in media for 14 days prior to crystal violet staining.

**Animal xenografts study and immunohistochemistry.** All experiments were performed under the approval of the Rutgers University Institutional Animal Care and Use Committee and complied with all relevant ethical regulations. Based on the experiment, 0.5, 1, 4.5 or $5\times10^6$ transduced NIH/3T3 cells were suspended in a 1:1 ratio in high concentration Matrigel basement membrane matrix (Corning) and injected subcutaneously in 6–8-week-old, female, athymic nude mice (Taconic, CrTac:NCr-Foxn1nu (NCr Nude) or NOD/SCID/interleukin 2 receptor γ null mice (The Jackson Laboratory, NOD.Cg-Prkdcscid Il2rgtm1Wjl /SzJ (NSG)). For tumor formation, $1\times10^6$ cells were injected for all three RET altered cell lines in nude mice. For RET$^{amp}$, tumor formation was further examined at $4.5\times10^6$ cells in NSG mice. Tumors were monitored twice weekly by palpation and tumor dimensions were measured using digital calipers (VWR). Tumor volumes were calculated using the formula ½(length × width$^2$). For drug treatment, $0.5\times10^6$ NCOA4-RET cells and $5\times10^6$ ΔRET and RET$^{amp}$ cells were injected bilaterally into flanks of athymic nude mice and were allowed to form palpable tumors before randomization. Allocation to treatment groups ($n = 5$ for NCOA4-RET and $n = 7$ for ΔRET and RET$^{amp}$ studies per group) was performed when average tumor volumes measured about 300 mm$^3$ using simple randomization with stratification and blinding to group allocation. Cabozantinib was dosed orally at 30 or 60 mg kg$^{-1}$ once daily for 14 days and was formulated according to the manufacturer's specifications. Tumor measurements and body weights were recorded three times a week for each animal upon initiating treatment. Immunohistochemical staining was prepared using a Ventana XT Discovery automated staining device according to the manufacturer's protocol. Primary antibodies Ki-67 (Spring Biosciences, M3062), Cleaved Caspase-3 (Cell Signaling Technology, 9661) were used at 1:400 and 1:200 dilutions respectively (Supplementary Table 4).

**Statistical analysis**. All experiments including three or more groups were analyzed using one-way ANOVA or two-way ANOVA for two variables. Multiple comparisons were performed using Tukey's test. All data represented as mean ± s.d. $p$ value ≤ 0.05 was considered statistically significant in all experiments. Analyses were performed using Prism, version 7.0d (GraphPad software).

**Reporting summary**. Further information on research design is available in the Nature Research Reporting Summary linked to this article.

## Data availability

The authors declare that the data supporting the findings in the study are available in the main article and supplementary files or available from corresponding authors upon request. A reporting summary for this article is available as a Supplementary Information file.

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

## Acknowledgements

The authors acknowledge the editorial, technical, and administrative assistance of Susan Moench, Jennifer Hostettler, and Jacqueline Harris. This work was supported by the AHEPA 5th District Cancer Research Foundation, the Stacy Goldstein Faculty Scholar Award established by Suzann and Edwin Goldstein (K.M.H.), National Institutes of Health Grant P30CA072720, through a generous gift to the Genetics Diagnostics to Cancer Treatment Program of the Rutgers Cancer Institute of New Jersey and Rutgers University Cell and DNA Repository (RUCDR) Infinite Biologics, Shared Resources of the Rutgers Cancer Institute of New Jersey including the Biospecimen Repository Service, Histopathology and Imaging, and Office for Human Research Service.

## Author contributions

K.M.H., B.L.-J., and S.M.A. conceived and supervised the study. B.S.P. designed, carried out experiments. B.S.P and J.H.C performed data analysis. W.P., K.P., S.L., and M.Y. carried out experiments. B.S.P., K.M.H. and J.H.C. co-wrote the manuscript. S.G. and A. K. advised on the study. B.L.-J., C.B.W., and K.W. gathered and contributed to clinical case data. A.B.S., L.M.G., F.D., and E.L. collected and analyzed data. J.S.R. and C.L. provided pathological analysis. B.Y. and J.H.C. conducted bioinformatics analysis. S.C.D., F.D., L.R.-R. and S.G. edited and revised the manuscript. All authors discussed the results and implications and commented on the manuscript at all stages.

## Additional information

**Competing interests:** J.H.C., B.Y., A.B.S., L.M.G., J.S.R., and S.M.A. are employees of and own stock in Foundation Medicine, Inc. K.M.H. is now an employee of and owns stock in Merck. The remaining authors declare no competing interests.

