## [Peer Review File · Nature Communications]

Reviewer #1 (Remarks to the Author):

By targeted genomic profiling, Authors describe for the first time, alterations of the RET receptor tyrosine kinase in a large cohort (n.9693) of breast cancers. RET gene rearrangements (n.16), missense mutations (n.25) and copy number amplifications (n.81) were found. Missense mutations were prevalently found in ER+ cases, while rearrangements/CNAs in TNBC cases. Two of the identified rearrangements, NCOA4-RET (previously identified and functionally studied in thyroid and lung cancers) and RASGEF1A-RET (named deltaRET, newly identified in this study), were able to promote proliferation and tumorigenicity of NIH373 fibroblasts. Multitargeted TKIs with anti-RET activity reduced viability of RET transduced MCF10A mammary epithelial cells and (cabozantinib) tumorigenicity of RET transduced NIH3T3 fibroblasts.

The study reports remarkably original, important and rigorous data, with highly relevant translational implications. In order to better detail their findings and derive more robust conclusions on the functional consequences of RET genomic alterations in breast cancer, Authors are requested to address the following points:

1. Based on the quoted literature, describing a role for RET in resistance to hormone therapy, it would be important to learn, whether in the studied cohort there is any correlation between response to hormone therapy and presence of RET genetic lesions.
2. Please report allele frequencies of RET missense mutations and rearranged introns within tumor samples, as well as proportion of tumor cells.
3. Some of the reported RET missense mutations are causally associated to mendelian MEN2 syndromes; have the Authors any information relative to the somatic vs germline nature of the RET missense mutations identified in breast cancers ?
4. Table 1 shows that the most common RET genomic alteration in breast cancer is copy number amplification. Importantly, this is virtually exclusive with respect to RET rearrangements/missense mutations. To further strengthen its relevance, it would be crucial to:
 - a) provide data relative to the minimal amplified region of chr. 10q (to address whether other genes might be involved in, e.g. whether RET is the only candidate driver of such a CNA).
 - b) provide data relative to the RET expression levels (mRNA) in the cases bearing this alteration.

c) detail whether the median copy number calculation was normalized for the abundance of cancer cells (see also point 2).

5. Fig. 2: effects of RETwt, NCOA4-RET and deltaRET on cell growth when expressed in MCF10A cells should be explored. This is important also to fully interpret response to TKI treatment (Fig. 4A).

6. In NIH3T3 cells, proliferative (Fig. 2B) and clonogenic (Fig. 2C) effects of deltaRET do not seem to strongly exceed those of RETwt. In addition, in NIH3T3 cells (Fig. 2D), phosphorylation level of deltaRET is difficult to be compared with that of RETwt because it is expressed at higher levels than RETwt. Finally, phosphorylation of deltaRET in MCF10A cells (Suppl. Fig 1) does not seem higher (considering levels of protein expression) than that of RETwt. Authors are asked to provide more quantitative comparisons between RETwt and deltaRET in terms of RET activity. Indeed (as commented on page 14, line 304), it remains still possible that one relevant mechanism conferring gain-of-function to deltaRET relies on its potentially altered expression level rather than ligand-independent potentiation of the intrinsic TK activity. In this frame, can the Authors determine RET mRNA levels in tissue samples bearing this rearrangement in comparison to samples negative for it (see also points 2 and 4)?

7. Fig. 2D: overall, levels of ERK and PI3K pathway activation from these Western blots do not convincingly demonstrate the activation of the RET signaling in the various transduced cells when compared to vector cells (see also point 6). Please address.

8. Fig. 4B: in deltaRET cells, decrease of RET phosphorylation upon cabozantinib treatment is not clearly detectable; moreover, drug treatment does not seem to strongly affect pAKT and pERK levels (pMEK might be a more accurate readout than pERK): please address.

9. While in NIH3T3 cell-based assays cabozantinib does not affect NCOA4-RET protein level (Suppl. 2A), it strongly reduces it in NIH3T3 xenografts (Fig. 5C). Delta-RET protein levels are similarly reduced by cabozantinib in NIH373 xenografts (Fig. 5C), but not in MCF10A cells (Fig. 5B). Please address.

10. Fig. 5C: to robustly establish effects of cabozantinib treatment on RET levels, phosphorylation and intracellular signal transduction, it would be necessary to analyze more than one single xenograft sample per each experimental point. Please address.

MINOR POINTS

1. The list of cancer-related genes and selected introns that were subjected to the NGS analysis should be reported.
2. Page 8, lines 175-9 and Fig 1C: clarify where does the predicted methionine start site map in the two RET fusions with intergenic sequences and whether the predicted ORFs are able to code for an intact RET TK domain.
3. Fig. 1C: alternative splicing of exons 19-21 of RET generate protein isoforms displaying different COOH-tails and using alternative STOP codons. Authors should predict whether or not tandem duplications at exon 20 are able to generate ORFs that are able to skip STOP codons and read through the fusion point.
4. Page 9, lines 201-206: tumorigenicity was assessed in nude mice with 1 million deltaRET or NCOA4-RET cells and in NSG mice with 4.5 million RETwt cells: detail this information also in the Methods section. Moreover, describe whether RETwt cells were tumorigenic when tested in the same conditions of the deltaRET and NCOA4-RET cells.
5. Page 10, line 214: cabozantinib is reported to be a Type II TKI (Zhao Z, et al Exploration of type II binding mode: A privileged approach for kinase inhibitor focused drug discovery? ACS Chem Biol. 2014 Jun 20;9(6):1230-41); please check.
6. Suppl. Fig. 3A and Fig. 5B: differently from what stated on page 11, lines 232-4, effects of cabozantinib on RETwt xenografts seem comparable to those on deltaRET ones; please check.
7. Fig. 6C: has patient response to the trastuzumab-cabozantinib-exemestane schedule to be attributed only to cabozantinib or can it depend also on the different hormone treatment?

Reviewer #2 (Remarks to the Author):

The manuscript describes analyses of the RET receptor tyrosine kinase in breast cancer. Authors begin with a screen of a very large panel of breast cancer patient samples (n=9693). This panel size is very interesting and could be discussed much further. Notably, RET mutations are very few in this sample. The authors focus on characterizing 2 rearrangements. An NCOA4-RET fusion protein also known as RET/PTC3 has been very well characterized previously in several cancers. The other rearrangement, involving RASGEF1A has not been previously described and is predicted to result in a truncated RET form using an alternative start in exon 11 (termed delta RET). The authors demonstrate that targeting of the NCOA4-RET with cabozantinib can be effective in vivo for managing patients with breast cancers bearing this mutation. Together, the data are convincing that the NCOA4-RET may act as an oncogene in breast. On the other hand, data for deltaRET are less strong, with inconsistent signaling, tumour growth and response to kinase inhibitors, making significance harder to interpret.

The authors provide a number of functional assays to compare the NCOA4-RET, deltaRET and wildtype RET. These data are very hard to compare as all three RET forms are frequently not performed in the same cell type or under the same conditions. Western blotting data shown are very concerning, as phosphorylation data are not consistent with the authors interpretation and positive and negative controls are generally missing. Statistical analyses are not always provided to support the interpretations and in some cases, better tests would be more appropriate (ANOVA vs t-tests).

Specific Comments

1] The text is overlong in some places and detail is missing in others, needing a balance adjustment. Details of the many possible ways to look for rearrangements (lines 80-92) and the extensive detail in the introduction and sections on rearrangements in the discussion could be simplified and summarized so that the authors could appropriately discuss their own data in more detail. The initial mutation screen data, for example, are under developed and could be usefully discussed in more detail.

2] The authors use t-tests to compare conditions in the majority of experiments. This allows them to compare 2 conditions but not to discuss variations amongst a group of conditions, which would necessitate different statistical tests (eg ANOVA). For example, lines 194-195 and figure 2c. There are large error bars for the deltaRET and WT RET conditions. They are each compared to vector control in fig 2 but not to each other so no conclusion should be drawn about their relative colony numbers without appropriate statistical comparison. There are a number of other similar examples in the text where the authors draw conclusions, which will need to be supported by the correct statistical comparison.

3] Phosphoprotein blots: In figure 2D there is no indication of variation in phosphor proteins under any conditions- this includes kinase dead mutant active proteins, wildtype RET and the truncation mutants and even the untransfected and empty vector control conditions. The authors do not discuss this, but it makes it very hard to interpret any of the remaining western data. Based on this figure, it appears that the RET constructs used have no effects on any signaling pathways in these cells. This is problematic for interpretation of the remaining experiments since these same models are used throughout. The authors should provide positive and negative controls for each of their remaining immunoblotting experiments for comparison if we are to be assured that RET is activating the pathways shown.

4] In Figure 4B, cabozantinib treatment reduces total NCOA4-RET protein, which would be very unusual. Please provide sufficient experimental detail so that these data can be interpreted. It looks from Fig 4B like it is simply loss of protein rather than protein inhibition that we are seeing. Cabozantinib clearly does not inhibit deltaRET well in this assay and does not affect downstream signals yet you saw some effects in vivo. This needs to be explained.

5] The mouse xenograft data are not comparable in the current format. The NCOA4 and deltaRET expressing cells were injected in different numbers from the wildtype. It appears that different mouse models were used for each of these experiments (athymic nudes versus NOD/SCID/IL-2R null mice) is this correct? Given these differences, the data cannot be compared and it would not be possible to draw conclusions about the relative growth of WTRET and truncation mutant cells (lines 201-206). Experiments comparing the WTRET and truncation mutants in the same animals under the same seeding conditions are essential or conclusions cannot be drawn here.

6] The in vivo tumour growth with cabozantinib treatment data are concerning. Data for NCOA4-RET is very nice with good growth and response curves. The growth curves for both deltaRET (5b) and WTRET (Supl 3) are much less convincing. Variability is clearly very large (error bars) for all conditions, but this is not discussed. The growth rates are very erratic, even for the saline treated conditions, increasing and decreasing (even the vehicle control) over the 14 days. It is very difficult to conclude that these data sets are comparable with the NCOA4 –RET results, as there appears to have been some difficulty with these specific assays.

7] Experimental detail is very limited and lacks specifics for many of the assays, making it harder to understand the outcomes (eg inhibitor treatment times, serum starvation times, antibody names). It would be helpful if the authors revisited their methods with the details requested in the Journal Check List as many of these points are not in the text and details are not provided.

Minor comments

- Scale bars or magnifications are needed in figures.
- Provide specifics of numbers of replicates in legends. Authors mention some assays were “repeated twice” which would not allow statistical comparisons. Clarify
- Provide appropriate statistical comparisons in figures- clearly indicate what conditions were compared.
- Figure 5- densitometry should appear in the main figure not the supplementary as the western blot shown is very difficult to interpret on its own.
- the crizotinib treatment in figure 4B is never mentioned or explained

We greatly appreciate the thorough review of our manuscript titled “*RET* rearrangements are actionable alterations in breast cancer ” and the thoughtful critique provided. We submit our revised manuscript and this response letter addressing each comment directly. These reviews have helped to significantly strengthen the manuscript and better clarify relevant details. In general, we have now added more details pertaining to the clinical cohort and nature of the study, new data, improved presentation and analysis of results, and made changes to address major and minor points as requested by the reviewers. The reviewer comments and revisions in the manuscript for each point are indicated below.

REVIEWER 1

By targeted genomic profiling, Authors describe for the first time, alterations of the RET receptor tyrosine kinase in a large cohort (n.9693) of breast cancers. RET gene rearrangements (n.16), missense mutations (n.25) and copy number amplifications (n.81) were found. Missense mutations were prevalently found in ER+ cases, while rearrangements/CNAs in TNBC cases. Two of the identified rearrangements, NCOA4-RET (previously identified and functionally studied in thyroid and lung cancers) and RASGEF1A-RET (named deltaRET, newly identified in this study), were able to promote proliferation and tumorigenicity of NIH3T3 fibroblasts. Multitargeted TKIs with anti-RET activity reduced viability of RET transduced MCF10A mammary epithelial cells and (cabozantinib) tumorigenicity of RET transduced NIH3T3 fibroblasts.

The study reports remarkably original, important and rigorous data, with highly relevant translational implications. In order to better detail their findings and derive more robust conclusions on the functional consequences of RET genomic alterations in breast cancer, Authors are requested to address the following points:

1. Based on the quoted literature, describing a role for RET in resistance to hormone therapy, it would be important to learn, whether in the studied cohort there is any correlation between response to hormone therapy and presence of RET genetic lesions.

We agree with the reviewer that it would also be highly relevant to determine if *RET* genomic alterations are correlated to hormone response. This point is also relevant for the clinical case presented in Fig.6 given the short interval of benefit with combined initial hormone and HER2-directed therapy. However, as it is not possible to tease out the contributions from *NCOA4-RET* alone, *HER2* amplification alone, or their functional combination from this case, these questions warrant further study and this is now stated in the text on page 14, lines 306-309.

Analysis of the *RET* genomic landscape has not been previously reported at this scale exclusively for breast cancer. Our cohort is a collection of breast cancers from different source institutions that were sequenced as part of routine clinical care. Treating physicians are not required to supply extensive diagnostic information or treatment history for cases when performing the assay, but this genomic panel is routinely used to inform treatment of advanced and/or refractory cancer cases. Apart from our own institutional cases, we are generally limited to tumor type, stage, and assay results for other cases. However, our cohort consists of ~60% advanced cases overall given that assayed tissues were primarily from sites beyond the breast (Table 1). Additionally, primary breast tumor may be assayed if distant site tumor tissue is insufficient either in quantity or quality, resulting in an underestimation of advanced cases in this cohort. Assuming that distant disease results after failure from primary treatment, including hormonal therapy, such a finding suggests that *RET* alterations may be found in hormone therapy-refractory disease. Whether *RET* alterations contributed to intrinsic or acquired resistance in individual cases cannot be answered with available data. We believe, however, that the frequent finding of *RET* alterations in our study across all breast cancer subtypes provides a significant and novel basis for assessing the relation between *RET* genetic lesions, expression levels, and treatment outcomes in those

selected patients for future studies. We have now highlighted this in our discussion and thank the reviewer for bringing attention to this.

2. Please report allele frequencies of RET missense mutations and rearranged introns within tumor samples, as well as proportion of tumor cells.

We appreciate the line of thought by reviewer 1 for this and the next two comments, particularly given the genomic assay is tumor only. Allele frequencies and tumor purity for *RET* missense mutations is now reported in a new Supplementary Table 4 and tumor purity for all *RET* alterations is now included in Supplementary Table 2. Unlike mutations, the assay is not validated to calculate allele frequencies for rearranged introns.

3. Some of the reported RET missense mutations are causally associated to mendelian MEN2 syndromes; have the Authors any information relative to the somatic vs germline nature of the RET missense mutations identified in breast cancers?

To further investigate the germline versus somatic nature of the missense mutations, we used a recently published approach that enables computational inference of somatic/germline status even in cases where matched normal tissues are not sequenced (Reference #24). We confirmed the status using a second algorithm for cases where ploidy was two (Reference #25). This information is now in the new Supplementary Table 4, reviewed in results on page 6, lines 125-131 and discussion on pages 13-14, lines 298-302, and references added in the main text:

24. Sun, J.X., *et al.* A computational approach to distinguish somatic vs. germline origin of genomic alterations from deep sequencing of cancer specimens without a matched normal. *PLoS computational biology* **14**, e1005965 (2018).
25. Khiabani, H., *et al.* Inference of germline mutational status and evaluation of loss of heterozygosity in high-depth tumor-only sequencing data. *Journal of Clinical Oncology Precision Oncology* (2017).

As presented in Supplementary Table 4, it is notable that some of the pathogenic variants are consistent with germline status and have also been associated with MEN syndromes with variable penetrance¹. At least one variant, M918T, has been reported to occur either as somatic or germline². In this cohort, family history and personal history of other cancers is not available. Identification of potentially germline *RET* mutations in a clinical setting, particularly when coupled with relevant personal and family history, would result in patient referral for genetic counseling and testing to evaluate their status further³. Of importance to treatment recommendations is that the previously reported *RET* V804M is a known gatekeeper mutation which contributes to cabozantinib resistance, but tumors with this mutation may still retain sensitivity to other tyrosine kinase inhibitors with activity against RET such as ponatinib and AD80 based on preclinical data⁴⁻⁶.

¹. Marquard J, Eng C. Multiple Endocrine Neoplasia Type 2, in *GeneReviews*® [Internet]. Adam MP, Ardinger HH, Pagon RA, et al., editors. Seattle (WA): University of Washington, Seattle; 1993-2018.

². Gimm O., *et al.* Over-representation of a germline RET sequence variant in patients with sporadic medullary thyroid carcinoma and somatic RET codon 918 mutation. *Oncogene* **18**, 1369-73 (1999).

³. Hirshfield KM., *et al.* Clinical Actionability of Comprehensive Genomic Profiling for Management of Rare or Refractory Cancers. *The oncologist* (2016).

⁴. Carlomagno, F., *et al.* Disease associated mutations at valine 804 in the RET receptor tyrosine kinase confer resistance to selective kinase inhibitors. *Oncogene* **23**, 6056-6063 (2004).

⁵. Mologni, L., *et al.* Ponatinib is a potent inhibitor of wild-type and drug-resistant gatekeeper mutant RET kinase. *Mol Cell Endocrinol* **377**, 1-6 (2013).

⁶. Plenker, D., *et al.* Drugging the catalytically inactive state of RET kinase in RET-rearranged tumors. *Science translational medicine* **9**(2017).

4. Table 1 shows that the most common RET genomic alteration in breast cancer is copy number amplification. Importantly, this is virtually exclusive with respect to RET rearrangements/missense mutations. To further strengthen its relevance, it would be crucial to:

- a) provide data relative to the minimal amplified region of chr.10q (to address whether other genes might be involved in, e.g. whether RET is the only candidate driver of such a CNA).**
- b) provide data relative to the RET expression levels (mRNA) in the cases bearing this alteration.**
- c) detail whether the median copy number calculation was normalized for the abundance of cancer cells (see also point 2).**

We agree that *RET* amplification is worthy of more in-depth follow up studies.

a) We recognize the reviewer’s concern for the presence of alternative oncogenes adjacent to *RET* and present in an amplicon on chromosome 10. Although *RET* itself is amplified and the sequencing assay covers *RET*, the targeted approach means that not all the surrounding chromosomal regions are captured. It is therefore not amenable to do such an analysis of the minimal area of the region with this assay. Notably, *RET* is the only compelling Tier 1 oncogene from the Cancer Gene Census on segment 10q11 (shown below). Additionally, a recent study demonstrated that induced overexpression of RET wildtype sequence in an inducible, transgenic mouse model resulted in luminal mammary tumors that were responsive to a RET kinase inhibitor (Reference 38).

38. Gattelli, A., *et al.* Chronic expression of wild-type Ret receptor in the mammary gland induces luminal tumors that are sensitive to Ret inhibition. *Oncogene* (2018).

Gene Symbol	Name	Entrez GeneId	Genome Location	Tier	Hallmark	Chr Band	Somatic	Germline	Role in Cancer
KLF6	Kruppel-like factor 6	1316	10:3779539-3785014	1		10p15	yes		TSG
NCOA4	nuclear receptor coactivator 4 - PTC3 (ELE1)	8031	10:51579142-51586417	1		10q11.2	yes		TSG; fusion
RET	ret proto-oncogene	5979	10:43077259-43128269	1		10q11.2	yes	yes	oncogene; fusion
CCDC6	coiled-coil domain containing 6	8030	10:59792917-59906424	1		10q21	yes		TSG; fusion

b) We do not have bioinformatics data on expression levels for *RET* amplifications as RNA sequences are analyzed for rearrangements alone. However, for one of the *RET* amplified breast cancers in this cohort from our own institution (* in Fig.1a), we were able to evaluate RET expression in several different tissues from this case. All tumor tissues or tumor cells from this case had detectable RET overexpression (now shown in Supplementary Fig.3c). We note that the correlation between *RET* amplification and RET overexpression may not hold true for all cases. We emphasize in the discussion that given the frequency of amplifications, its functional relevance is crucial to explore.

c) Copy number is corrected for tumor purity and the method of detection is elaborated below and published in the following article. This is now clarified in methods and the reference is included:

63. Frampton, G.M., *et al.* Development and validation of a clinical cancer genomic profiling test based on massively parallel DNA sequencing. *Nature biotechnology* **31**, 1023-1031 (2013).

CNA detection. Using a comparative genomic hybridization (CGH)-like method, we obtained a log-ratio profile of the sample by normalizing the sequence coverage obtained at all exons and ~3,500 genome-wide SNPs against a process-matched normal control. This profile was corrected for GC-bias, segmented and interpreted using allele frequencies of sequenced SNPs to estimate tumor purity and copy number at each segment. Briefly, if S_i is a genomic segment at constant copy number in the tumor, let l_i be the length of S_i , r_{ij} be the coverage measurement of exon j within S_i , and f_{ik} be the minor allele frequency of SNP k within S_i . We estimate p , tumor purity, and C_i , the copy numbers of S_i . We jointly model r_{ij} and f_{ik} , given p and C_i :

$$r_{ij} \sim N\left(\log_2 \frac{p * C_i + (1 - p) * 2}{p * (\sum_i l_i C_i) / \sum_i l_i + (1 - p) * 2}, \sigma_{r_{ij}}\right)$$

and,

$$f_{ik} \sim N\left(\frac{p * M_i + (1 - p)}{p * C_i + (1 - p) * 2}, \sigma_{f_{ik}}\right)$$

where M_i is the copy number of minor alleles at S_i , distributed as integer $0 \leq M_i \leq C_i$. $\sigma_{r_{ij}}$ and $\sigma_{f_{ik}}$ reflect noise observed in the CGH and SNP data, respectively. Fitting was performed using Gibbs sampling, assigning absolute copy number to all segments. Model quality was reviewed and alternative explanations considered, and focal amplifications are called at segments with ≥ 6 copies (or ≥ 7 for triploid; ≥ 8 for tetraploid tumors) and homozygous deletions at 0 copies, in samples with purity $> 20\%$.

5. Fig. 2: effects of RETwt, NCOA4-RET and delta RET on cell growth when expressed in MCF10A cells should be explored. This is important also to fully interpret response to TKI treatment (Fig. 4A).

We agree that effects of *RET* alterations on growth of MCF10A cells is important to accurately analyze TKI treatment response in Fig.4a. The growth effect is now included as part of Supplementary Fig.1a. This data shows that while there are no differences at day 4, we observe a significant increase in cell numbers for *RET* wildtype, Δ *RET*, and NCOA4-*RET* on day 8 when compared to vector. For the inhibitor assay, cells are plated and then treated with drug for 72h, after which the response is evaluated. The time duration for this drug response assay matches day 4 to day 6 on the growth curve. The absence of growth differences at this time frame suggests that growth rates do not confound the differences observed due to TKI treatment. This is now described on page 8, lines 166-168 and page 9, lines 198-200.

6. In NIH3T3 cells, proliferative (Fig. 2B) and clonogenic (Fig. 2C) effects of deltaRET do not seem to strongly exceed those of RETwt. In addition, in NIH3T3 cells (Fig. 2D), phosphorylation level of deltaRET is difficult to be compared with that of RETwt because it is expressed at higher levels than RETwt. Finally, phosphorylation of deltaRET in MCF10A cells (Suppl. Fig 1) does not seem higher (considering levels of protein expression) than that of RETwt. Authors are asked to provide more quantitative comparisons between RETwt and deltaRET in terms of RET activity. Indeed (as commented on page 14, line 304), it remains still possible that one relevant mechanism conferring gain-of-function to deltaRET relies on its potentially altered expression level rather than ligand-independent potentiation of the intrinsic TK activity. In this frame, can the Authors determine RET mRNA levels in tissue samples bearing this rearrangement in comparison to samples negative for it (see also points 2 and 4)?

We recognize that our description of this data in the initial submission was not optimal. We agree that it may confuse the readers in terms of the comparisons and conclusions derived between Δ RET and RET wt. Both parental NIH3T3 and MCF10A cells do not express RET at levels detectable by western blot (as shown for parental and vector lanes in Fig. 2a). We use an exogenous expression model where cell lines are engineered to express the relevant RET protein at a level that is higher level than in the endogenous setting, including RET wt. This creates confusion in the assessment, since the RET wt here does not represent endogenous RET wildtype levels of the parental cells but represents the full-length, wildtype sequence of RET which is overexpressed and serves to model *RET* amplification. We have renamed RETwt as RET^{amp} to reflect this representation.

RET amplification is hypothesized to also activate RET kinase since copy number amplification leading to aberrant overexpression of RTKs is a known mechanism for kinase activation, such as for HER2. In this context, Δ RET may not necessarily be more active than RETwt overexpression itself. The comparison for Fig.2d is to test which of these alterations (overexpression representing active *RET* amplification, Δ RET, and NCOA4-RET) exhibits kinase activation as evaluated by phosphorylation in the absence of serum or growth factor stimulation. Based on the detection of phosphorylation in the absence of stimulation (more specifically in the case of NIH 3T3, also in the absence of expression of GFR α , the co-receptor required for ligand binding and receptor activation), we report that each of the alterations is constitutively active. The negative and positive controls for this assay are RET K758M (kinase dead mutant) and RET M918T (constitutively active kinase mutant), respectively.

For clarification, we have modified Figure 2. We have revised labeling of RET wt to RET^{amp} throughout the manuscript as it serves to model an active amplification with altered and increased level of expression. We have included quantitative comparisons between all the RET alterations for activity in Fig. 2d and relative levels to the true negative control in this case, which is the kinase inactive RET K758M. As the reviewer has accurately identified, it is also possible that Δ RET may rely on altered expression level. Unfortunately, we do not have access to confirm mRNA levels for this in the patient tissue since available tissue was from the patient's early stage disease with sparingly little formalin fixed paraffin embedded tissue that was processed for targeted intron captured sequencing. The RNA sequencing data confirms the presence of manual reads for the rearrangement.

7. Fig. 2D: overall, levels of ERK and PI3K pathway activation from these Western blots do not convincingly demonstrate the activation of the RET signaling in the various transduced cells when compared to vector cells (see also point 6). Please address.

Due to the complexity of the earlier figure 2D and its difficult interpretation, we both optimized the experiment and simplified Fig.2D to only present RET kinase activation. In order to address downstream signaling, we have optimized the experiment and have now included the downstream signaling blots in Fig.2E with appropriate controls. In this context of activation, pMEK (as suggested by this reviewer in comment #8) was a better readout of the MAPK pathway and similarly, pS6 for the PI3K/AKT/mTOR pathway. For Fig.2E, we have also included the corresponding quantified signal intensities of multiple independent experiments with appropriate statistical analysis to better clarify the downstream effects of these alterations.

8. Fig. 4B: in deltaRET cells, decrease of RET phosphorylation upon cabozantinib treatment is not clearly detectable; moreover, drug treatment does not seem to strongly affect pAKT and pERK levels (pMEK might be a more accurate readout than pERK): please address.

We agree that the difference in Fig.4B for deltaRET was not clearly detectable. In the revised manuscript, we have added a better representation of this result and quantified the image density data which is now included in the main figure with appropriate statistical analysis.

9. While in NIH3T3 cell-based assays cabozantinib does not affect NCOA4-RET protein level (Suppl. 2A), it strongly reduces it in NIH3T3 xenografts (Fig. 5C). Delta-RET protein levels are similarly reduced by cabozantinib in NIH373 xenografts (Fig. 5C), but not in MCF10A cells (Fig. 4B). Please address.

Fig.5c (now includes n=3 per treatment condition) is a measurement on tumor protein that was collected from tumor remaining after 2 weeks of treatment in xenograft animals. As shown in Fig. 5a and 5b, and in 5d by histology and IHC images, tumor volume and tumor cell density is significantly reduced in treatment groups compared to vehicle. This clearance of tumor cells in turn, results in reduction in the absolute volume of tumor cells expressing the fusion protein, which is confirmed in Fig. 5c.

In Supplementary Fig. 2a and Fig. 4b, measurements are made after treating the cells with cabozantinib for 1 hour. This is to probe the effect of cabozantinib on inhibition of the NCOA4-RET and Δ RET kinases, which is the mechanism of action for the Type II TKI. Therefore, while we see decreases in phospho-protein, we do not see similar changes in total protein levels.

10. Fig. 5C: to robustly establish effects of cabozantinib treatment on RET levels, phosphorylation and intracellular signal transduction, it would be necessary to analyze more than one single xenograft sample per each experimental point. Please address.

We have modified Fig.5c to include analysis of three xenograft samples per treatment condition, quantitative analysis of protein, signaling changes, and appropriate statistical analyses.

MINOR POINTS

1. The list of cancer-related genes and selected introns that were subjected to the NGS analysis should be reported.

We have now included the targeted gene panels as Supplementary Table 1.

2. Page 8, lines 175-9 and Fig 1C: clarify where does the predicted methionine start site map in the two RET fusions with intergenic sequences and whether the predicted ORFs are able to code for an intact RET TK domain.

As a frame of reference, the predicted start site of Δ RET maps to methionine at residue 674 in exon 11. As the kinase domain starts at residue 724, Δ RET codes for an intact TK domain. For *RET* exons involved in the two *RET* rearrangements with intergenic sequences (Fig.1c), the predicted start site would map to methionine at residue 759 in exon 12 and is able to code for the remaining kinase domain. Intergenic rearrangements may also be truncation mutants of the *RET* wildtype kinase domain sequence, similar to Δ RET, but with the absence of more N-terminal residues. But unlike Δ RET, which is the product of a rearrangement between *RASGEF1A* and *RET*, these rearrangements are within an intergenic space. Therefore, there is the potential for an alternate start site mapping prior to the inclusive *RET* exons. In this case, the rearrangement may include the intact *RET* TK domain as exon 12 starts at residue 712. This is now described on page 7, lines 149-154.

3. Fig. 1C: alternative splicing of exons 19-21 of RET generate protein isoforms displaying different COOH-tails and using alternative STOP codons. Authors should predict whether or not tandem duplications at exon 20 are able to generate ORFs that are able to skip STOP codons and read through the fusion point.

The reviewer makes a very interesting comment. Predicting ORFs with duplicated exons does not detect potential alternative splicing that could lead to kinase domain duplications as these algorithms recognize

the native STOP codons. The recurrent finding of such duplications warrants further detailed enquiry including intronic sequences and scoring of potential splice regulator sites using validated splice prediction tools. There is a precedent for kinase domain duplications, which have been reported in other kinases such as EGFR, MET, BRAF, FGFR, including their subsequent functionality and therapeutic actionability. This is now discussed on page 15, lines 336-344 and the following references are included:

41. Baik, C.S., Wu, D., Smith, C., Martins, R.G. & Pritchard, C.C. Durable Response to Tyrosine Kinase Inhibitor Therapy in a Lung Cancer Patient Harboring Epidermal Growth Factor Receptor Tandem Kinase Domain Duplication. *Journal of thoracic oncology : official publication of the International Association for the Study of Lung Cancer* **10**, e97-99 (2015).
42. Klemptner, S.J., *et al.* Identification of BRAF Kinase Domain Duplications Across Multiple Tumor Types and Response to RAF Inhibitor Therapy. *JAMA oncology* **2**, 272-274 (2016).
43. Plenker, D., *et al.* Structural alterations of MET trigger response to MET kinase inhibition in lung adenocarcinoma patients. *Clinical Cancer Research* (2017).

4. Page 9, lines 201-206: tumorigenicity was assessed in nude mice with 1 million deltaRET or NCOA4-RET cells and in NSG mice with 4.5 million RETwt cells: detail this information also in the Methods section. Moreover, describe whether RETwt cells were tumorigenic when tested in the same conditions of the deltaRET and NCOA4-RET cells.

We apologize for confusion generated as this data was not presented previously: when wildtype RET (RET^{amp}) was injected into athymic nude mice at the same number as with NCOA4-RET and ΔRET in Fig.3a (1*10⁶), we did not observe any tumor formation. This is important to describe and we have now added this observation to Fig.3a results. As a separate investigation into the effect of RET amplification, and in order to assess if a greater cell number might result in tumor formation, we attempted 4.5*10⁶ cells and in NOD/SCID/interleukin 2-receptor γ null mice since they are more immunodeficient in comparison to athymic nudes. These results are described in Fig.3b. The current Fig.3a compares all three RET alterations at the same cell number in the same mouse model. Text for results, legends, and methods has been modified accordingly.

5. Page 10, line 214: cabozantinib is reported to be a Type II TKI (Zhao Z, et al Exploration of type II binding mode: A privileged approach for kinase inhibitor focused drug discovery? ACS Chem Biol. 2014 Jun 20;9 (6):1230-41); please check.

We thank the reviewer for identifying this error (also confirmed with references below), which has been corrected in the manuscript appropriately.

Li, M.-J., *et al.* Development of efficient docking strategies and structure-activity relationship study of the c-Met type II inhibitors. *Journal of Molecular Graphics and Modeling* **75**, 241-249 (2017).

Bahcall, M., *et al.* Acquired METD1228V Mutation and Resistance to MET Inhibition in Lung Cancer. *Cancer Discovery* **6**, 1334-1341 (2016).

6. Suppl. Fig. 3A and Fig. 5B: differently from what stated on page 11, lines 232-4, effects of cabozantinib on RETwt xenografts seem comparable to those on deltaRET ones; please check.

Upon using the appropriate statistical analysis for the entire growth curve (two-way ANOVA for all time points and multiple comparisons tests between all groups), we have now clarified this statement to read “RET^{amp} xenografts also showed significant reduction in tumor volumes with both doses of cabozantinib in comparison to vehicle.... (Supplementary Fig.3a and 3b).” on page 10, lines 226-227.

7. Fig. 6C: has patient response to the trastuzumab-cabozantinib-exemestane schedule to be attributed only to cabozantinib or can it depend also on the different hormone treatment?

The reviewer brings up an interesting point, which is highly relevant. In this case, whether the response was due to a cabozantinib, a different hormone treatment, or the combination of cabozantinib with the new hormone treatment cannot be assessed with certainty since both drugs were initiated at the same time. Similar to the effect of RET inhibitors in combination with aromatase inhibitors for ER+ cancers, further studies in RET rearranged cases will be needed to tease out these differences and investigate the role of RET inhibitor alone. It may well be that combinatorial therapy with RET inhibitors will be of optimal benefit but further studies are needed to make this determination. This statement has now been added to the results for Fig.6.

REVIEWER 2

The manuscript describes analyses of the RET receptor tyrosine kinase in breast cancer. Authors begin with a screen of a very large panel of breast cancer patient samples (n=9693). This panel size is very interesting and could be discussed much further. Notably, RET mutations are very few in this sample. The authors focus on characterizing 2 rearrangements. An NCOA4-RET fusion protein also known as RET/PTC3 has been very well characterized previously in several cancers. The other rearrangement, involving RASGEF1A has not been previously described and is predicted to result in a truncated RET form using an alternative start in exon 11 (termed delta RET). The authors demonstrate that targeting of the NCOA4-RET with cabozantinib can be effective in vivo for managing patients with breast cancers bearing this mutation. Together, the data are convincing that the NCOA4-RET may act as an oncogene in breast. On the other hand, data for deltaRET are less strong, with inconsistent signaling, tumor growth and response to kinase inhibitors, making significance harder to interpret.

The authors provide a number of functional assays to compare the NCOA4-RET, deltaRET and wildtype RET. These data are very hard to compare as all three RET forms are frequently not performed in the same cell type or under the same conditions. Western blotting data shown are very concerning, as phosphorylation data are not consistent with the authors interpretation and positive and negative controls are generally missing. Statistical analyses are not always provided to support the interpretations and in some cases, better tests would be more appropriate (ANOVA vs t-tests).

Specific Comments

1] The text is overlong in some places and detail is missing in others, needing a balance adjustment. Details of the many possible ways to look for rearrangements (lines 80-92) and the extensive detail in the introduction and sections on rearrangements in the discussion could be simplified and summarized so that the authors could appropriately discuss their own data in more detail. The initial mutation screen data, for example, are under developed and could be usefully discussed in more detail.

We thank reviewer 2 for this valuable input as a more concise introduction has allowed us to develop methods, results, and discussion in greater detail. This has alleviated space constraints in addressing subsequent concerns of both reviewers and clarifying results. We have replaced the prior introduction and discussion with revised verbiage to reflect new/updated data as well as points raised by both reviewers.

2] The authors use t-tests to compare conditions in the majority of experiments. This allows them to compare 2 conditions but not to discuss variations amongst a group of conditions, which would necessitate different statistical tests (eg ANOVA). For example, lines 194-195 and figure 2c. There

are large error bars for the deltaRET and WT RET conditions. They are each compared to vector control in fig 2 but not to each other so no conclusion should be drawn about their relative colony numbers without appropriate statistical comparison. There are a number of other similar examples in the text where the authors draw conclusions, which will need to be supported by the correct statistical comparison.

We have now revisited all of our statistical comparisons using appropriate tests to compare multiple groups, which are now elaborated in methods and in each figure legend. As most figures use more than three groups, ANOVA tests have been applied where required. All conclusions drawn in the paper are supported by appropriate statistical analysis.

3] Phosphoprotein blots: In figure 2D there is no indication of variation in phosphor proteins under any conditions- this includes kinase dead mutant active proteins, wildtype RET and the truncation mutants and even the untransfected and empty vector control conditions. The authors do not discuss this, but it makes it very hard to interpret any of the remaining western data. Based on this figure, it appears that the RET constructs used have no effects on any signaling pathways in these cells. This is problematic for interpretation of the remaining experiments since these same models are used throughout. The authors should provide positive and negative controls for each of their remaining immunoblotting experiments for comparison if we are to be assured that RETS is activating the pathways shown.

We recognize that our description and presentation of this data was not optimal in the original manuscript. We have resolved this by making clarifications for the experiments in Fig.2. We have modified labeling of RET wt to RET^{amp} throughout the manuscript as this engineered cell line serves to model an active amplification with increased level of expression. We agree with the reviewer that the data, as previously presented, did not convincingly demonstrate differences in signaling. Due to the complexity of the original Fig.2d, we both optimized the experiment and simplified Fig.2d to only present RET kinase activation. In order to address downstream signaling, we have optimized the experiment, have now included the downstream signaling blots in Fig.2e with appropriate controls.

For both Fig.2d and Fig.2e, we have included the vector control, kinase inactive mutant RET K758M (negative control), and constitutively active kinase mutant RET M918T (positive control). In the new Fig.2d, the quantitative comparisons for activation of RET kinase are made relative to RET K758M and are included in the main figure. In the context of downstream activation, pMEK (as suggested by reviewer 1 in comment #8) was a better readout of the MAPK pathway and similarly, pS6 for the PI3K pathway. For Fig.2E, we have also included quantified signal intensities of multiple independent experiments with appropriate statistical analysis to clarify the downstream effects of these alterations. Here, comparisons are made relative to vector control. We have now elaborated and clarified the methods, legend, and results for Fig.2d and Fig.2e.

4] In Figure 4B, cabozantinib treatment reduces total NCOA4-RET protein, which would be very unusual. Please provide sufficient experimental detail so that these data can be interpreted. It looks from Fig 4B like it is simply loss of protein rather than protein inhibition that we are seeing. Cabozantinib clearly does not inhibit deltaRET well in this assay and does not affect downstream signals yet you saw some effects in vivo. This needs to be explained.

Fig. 4b has now been modified to better represent the effect of the inhibitor, along with quantitative comparisons. We agree that the difference in Fig.4b for Δ RET was not clearly detectable previously. We suspect that this was due to the method in which the experiment was performed, where, transfection of cells was done in individual wells and then exposed to increasing concentrations of the inhibitor. Alternatively, we transfected a pool of cells, reseeded them into different wells and then incubated with

inhibitor at increasing concentrations. Due to less variability with the alternative approach, we can now visually see a reduction in phospho-proteins relative to total protein levels in the revised figure for both NCOA4-RET and Δ RET. Concomitantly, reduction in downstream MAPK signaling is also observed. The quantification of signal intensities from three independent experiments support statistically significant decrease in phospho/total levels with increasing inhibitor concentrations suggesting inhibition of the kinase activity. Although there is a decrease in tRET compared to control, especially for NCOA4-RET, it is now verified that the reduction in pRET is not due to total levels alone, since there is a significant decrease even after normalization to total RET levels. Changes in tRET with the use of RET inhibitors have been observed by others (reference 27) and we conjecture that they may be explained by indirect or off target effects of these multi-target kinase inhibitors. This explanation is now included on pages 9-10, lines 205-208.

27. Mologni, L., Redaelli, S., Morandi, A., Plaza-Menacho, I. & Gambacorti-Passerini, C. Ponatinib is a potent inhibitor of wild-type and drug-resistant gatekeeper mutant RET kinase. *Molecular and cellular endocrinology* **377**, 1-6 (2013).

5] The mouse xenograft data are not comparable in the current format. The NCOA4 and deltaRET expressing cells were injected in different numbers from the wildtype. It appears that different mouse models were used for each of these experiments (athymic nudes versus NOD/SCID/IL-2R null mice) is this correct? Given these differences, the data cannot be compared and it would not be possible to draw conclusions about the relative growth of WTRET and truncation mutant cells (lines 201-206). Experiments comparing the WTRET and truncation mutants in the same animals under the same seeding conditions are essential or conclusions cannot be drawn here.

We apologize for the confusion generated as this data was not presented previously - when wildtype RET (RET^{amp}) was injected into athymic nude mice at the same number as in NCOA4-RET and Δ RET in Fig.3a (1×10^6), we did not observe any tumor formation. This is important to describe and we have now added this observation to Fig.3a results (page 8, lines 179-182). As a separate investigation into the effect of RET amplification, and in order to assess if a greater cell number might result in tumor formation under more permissible conditions, we attempted 4.5×10^6 cells and in NOD/SCID/interleukin 2-receptor γ null mice since they are more immunodeficient in comparison to athymic nudes. These results are described in Fig.3b. The current Fig.3a compares all three RET alterations at the same cell number in the same mouse model. Pertinent text in the results, legends, and methods has been modified accordingly.

6] The in vivo tumor growth with cabozantinib treatment data are concerning. Data for NCOA4-RET is very nice with good growth and response curves. The growth curves for both deltaRET (5b) and WTRET (Supl 3) are much less convincing. Variability is clearly very large (error bars) for all conditions, but this is not discussed. The growth rates are very erratic, even for the saline treated conditions, increasing and decreasing (even the vehicle control) over the 14 days. It is very difficult to conclude that these data sets are comparable with the NCOA4 –RET results, as there appears to have been some difficulty with these specific assays.

With our pilot experiments, we were able to assess that NCOA4-RET was the fastest growing xenograft and that 0.5×10^6 cells were sufficient to induce tumor formation within a week. There was also less variability with tumor formation and response to cabozantinib for NCOA4-RET, which is evident from the error bars and gross tumor images. In case of Δ RET, we suspected that increasing the cell number might reduce the latency of tumor formation. We also observed some degree of asymmetric volume reduction in tumors in response to cabozantinib in our pilots and these experiments that could have resulted in measurement variability. Since the goal was to test if each of these RET alterations was sensitive to cabozantinib, but not necessarily to compare between them, we used a different cell number for both Δ RET and RET^{amp} experiments in order to offset the time for tumor formation. This too could

have contributed to the increased variability in tumor volumes between mice. Anticipating this may occur and based on pilot experiments, we increased the sample size to ensure that we would have enough power to make a statistically sound comparison between vehicle and treatment groups. In each individual experiment, the aim was to determine if the tumors respond to cabozantinib and if we observe statistically significant differences for both doses of cabozantinib as compared to vehicle. We have now included multiple comparison assessments using ANOVA in Fig.5 a, b and supplementary Fig. 3a as suggested by the reviewer. In the case of RET^{amp} as shown in Supplementary Fig.3b, we also notice variability in v5 protein expression within each group, which could also explain the larger variability and error bars for tumor volume in Supplementary Fig. 3a. More descriptive methods and results were added to all relevant sections to address these concerns.

7] Experimental detail is very limited and lacks specifics for many of the assays, making it harder to understand the outcomes (eg inhibitor treatment times, serum starvation times, and antibody names). It would be helpful if the authors revisited their methods with the details requested in the Journal Check List as many of these points are not in the text and details are not provided.

We agree with the reviewer that the initial manuscript had insufficient details as noted above. We have made substantial changes in the manuscript to address these limitations, updated the Journal Check List, and paid close attention to each detail and method.

Minor comments

- **Scale bars or magnifications are needed in figures.**

These have now been included in all figures where appropriate.

- **Provide specifics of numbers of replicates in legends. Authors mention some assays were “repeated twice” which would not allow statistical comparisons.**

To clarify, we have incorporated a specific ‘n’ in all legends and modified the earlier description where ‘repeated twice’ was used (this was in the context of repeated twice beyond the representative results in the figure for a total of 3). Where representative results are presented, we have verified the total number of times the experiments were performed.

- **Provide appropriate statistical comparisons in figures- clearly indicate what conditions were compared.**

We have now included information on conditions, appropriate statistics, and comparisons in all legends and figures. Figures have relevant brackets and connectors to mark the groups that are being compared with p-values and level of significance defined in figure legends.

- **Figure 5- densitometry should appear in the main figure not the supplementary as the western blot shown is very difficult to interpret on its own.**

To improve ease of data interpretation, the relevant densitometry charts have been moved to the main Fig.5c in the revised manuscript. We have also included 3 xenograft samples per treatment condition in the western blot and in the image density analysis.

- **the crizotinib treatment in figure 4B is never mentioned or explained**

Fig. 4b is now modified based on reviewer comments. Crizotinib has now been removed since it is not relevant to the results described here.

Reviewer #1 (Remarks to the Author):

The study is original and important for its basic and translational implications. This revision is strongly improved with respect to the original submission. In my opinion there remain a few points requiring Authors attention:

1. Demonstration and quantification of signaling of the various RET mutants has significantly improved with respect to the original submission. Still, as far as effects of RET inhibition by cabozantinib in NCOA4-RET/NIH3T3 fibroblasts, Suppl. Fig. 2 shows pAKT and pERK as readouts of drug activity, while Fig 2D reports pS6 (and not pMEK) as a readout of NCOA4-RET activity with respect to negative controls (vector and kinase-dead mutant). Authors are asked to match these markers. As an example, they may show in Suppl Fig. 2 also pAKT and pERK levels in untreated control cells to allow to figure-out to what extent cabozantinib effects on these two signaling mediators in NCOA4-RET cells are indeed mediated by NCOA4-RET blockade.

2. Page 2, line 1. Prevalence of RET fusions in PTC depends on age and environmental factors (e.g. highest in pediatric population and upon ionizing radiation exposure). Perhaps, here Authors would better report the prevalence in the general population as it is reported in the high-throughput screen of the TCGA study (Cell. 2014 Oct 23;159(3):676-90. doi:

10.1016/j.cell.2014.09.050)

3. Please specify in Methods or Figure legends whether growth/cell viability assays of Fig 2B (NIH3T3), Fig S1A (MCF10A), and Fig 4a (MCF10A) were performed in full serum or reduced serum concentration

Reviewer #2 (Remarks to the Author):

This is a resubmission of a previous manuscript on RET rearrangements in breast cancer. The authors have attempted to address many of the reviewer's previous comments, which was appreciated.

There remain a number of concerns about the data and the presentation thereof.

1] The use of the two in vitro cell models is important, but it is clear that expression of the two RET rearrangements had somewhat different effects in these 2 cell types. The authors bounce back and

for the with NIH 3T3 in the text and MCF10A in the supplementary data and then the reverse and then back again. The authors really need to be consistent, following one or both cell models so that the reader can compare figures and outcomes of the experiments.

2] There remain a number of concerns about the phosphorylation data in western blots in several figures. In figure 2E, the data are not comparable to any other figure panels. The effects of NCOA4_RET on pMEK is very surprising (no effect) which raises concerns on the effects on ERK in subsequent figures. In figure 4b, there does not really look to be an effect of cabozantinib on ERK phosphorylation, particularly for deltaRET. If the relatively reduced level of total RET in these lanes is taken into account- it looks like the amount of pERK would not be decreased significantly. It suggests the ERK activation they are seeing might not be RET dependent in this model as it isn't affected by cabozantinib.

3] The mouse tumor growth models remain hard to compare. The authors use 10 X more cells for the NCOA4-RET model compared to the delta RET and WTRET, and only ½ the number of animals (Figure 5 legend). Just about all you can say about these data is that NCOA4-RET is good at making tumors and the other constructs are less consistently so. The tumor sizes for delta RET are quite variable (Figure 5B) and this make the data in figure 5C difficult to rationalize.

4] It seems on looking through the data, that the authors have identified an oncogenic RET form, the NCOA4-RET, but that the delta RET may not be a very oncogenic form, if at all. It is not a very strong promoter of proliferation or colony formation compared to the WTRET form (Fig 2B). It doesn't induce pMEK or p70S6 more strongly (Fig 2E) and is significantly less able to induce pAKT or pERK than WTRET. It is a very slow inducer of tumor growth (Fig 3A, 5B). It's response to cabozantinib is quite inconsistent (Fig 5bcd, SupFig 2C). Perhaps the authors should consider moving the discussion of stronger and weakly transforming rearrangements up front in their manuscript and discuss the two variants from that perspective throughout. It would make the data easier to compare and would clearly highlight the differences between their models, which is an important point for discussion.

Additional comments

Lines 80-81 different numbers of RET alterations mentioned in these 2 lines (121 and 122)

In figure 2E The authors use pMEK and p70S6 to compare the activity of all their RET constructs, but all other figures and assays compare activity of ERK and AKT- The authors should replace this panel with comparable data to that in the other figures.

We are grateful for the opportunity to provide a second revision of our manuscript titled "*RET* rearrangements are actionable alterations in breast cancer". The original 2 referees raised important points as well as nuances in the data presentation and interpretation that we have addressed in the manuscript and below.

Reviewer #1 (Remarks to the Author):

The study is original and important for its basic and translational implications. This revision is strongly improved with respect to the original submission. In my opinion there remain a few points requiring Authors attention:

1. Demonstration and quantification of signaling of the various RET mutants has significantly improved with respect to the original submission. Still, as far as effects of RET inhibition by cabozantinib in NCOA4RET/ NIH3T3 fibroblasts, Suppl. Fig. 2 shows pAKT and pERK as readouts of drug activity, while Fig 2D reports pS6 (and not pMEK) as a readout of NCOA4RET activity with respect to negative controls (vector and kinase dead mutant). Authors are asked to match these markers. As an example, they may show in Suppl Fig. 2 also pAKT and pERK levels in untreated control cells to allow to figure out to what extent cabozantinib effects on these two signaling mediators in NCOA4-RET cells are indeed mediated by NCOA4-RET blockade.

We thank the reviewer for the positive feedback acknowledging improvements in the first revision. In accordance with both reviewers, to maintain consistency, we have revised all relevant figures to show p-P70 S6/P70 S6 and p-MEK/MEK as the downstream markers for RET signaling and drug activity.

For **Supplementary Fig. 3a** (previously Supplementary Fig. 2a), we have included results from vector control cells to compare the effect of cabozantinib in inhibiting signaling mediators downstream of RET (page 9, lines 192-194). We find that, under the same conditions, cabozantinib inhibits phosphorylation of P70 S6 and MEK effectively in NCOA4-RET cells but not in vector cells which do not express detectable levels of RET. These results are consistent with inhibition of downstream signaling being RET fusion-specific rather than due to off-target effects.

2. Page 2, line 1. Prevalence of RET fusions in PTC depends on age and environmental factors (e.g. highest in pediatric population and upon ionizing radiation exposure). Perhaps, here Authors would better report the prevalence in the general population as it is reported in the high throughput screen of the TCGA study (Cell. 2014 Oct 23;159 (3):67690. doi: 10.1016/j.cell.2014.09.050)

We appreciate this correction on frequency in PTC cases. As identified by the reviewer, the manuscript now reports the most recent frequency based on high throughput TCGA analysis on page 2, lines 24-25. The suggested reference is included accordingly (reference # 1, page 26, line 577).

3. Please specify in Methods or Figure legends whether growth/cell viability assays of Fig 2B (NIH3T3), Fig S1A (MCF10A), and Fig 4a (MCF10A) were performed in full serum or reduced serum concentration

Growth/cell viability assays were performed under full serum culture conditions as recommended by ATCC (American Type Culture Collection) for both cell lines. The detail on serum condition for assays is now included and clarified in the methods section: "Cell lines and *in vitro* overexpression of ORFs", on page 22, lines 493-497. Apart from experiments that involved measurement of phosphorylated proteins for signaling analysis, all other assays were performed under full serum conditions as recommended by ATCC for the cell line. Figure legends also include serum conditions when relevant.

Reviewer #2 (Remarks to the Author):

This is a resubmission of a previous manuscript on RET rearrangements in breast cancer. The authors have attempted to address many of the reviewer's previous comments, which was appreciated.

There remain a number of concerns about the data and the presentation thereof.

1] The use of the two in vitro cell models is important, but it is clear that expression of the two RET rearrangements had somewhat different effects in these 2 cell types. The authors bounce back and forth with NIH 3T3 in the text and MCF10A in the supplementary data and then the reverse and then back again. The authors really need to be consistent, following one or both cell models so that the reader can compare figures and outcomes of the experiments.

We thank the reviewer for this input on the previous submission to improve the clarity of this resubmission. We agree that data from two cell line models is important and maintain data from both in the manuscript. However, we have now revised the main figures to show data using NIH/3T3 cells. All data pertaining to the MCF10A cells has been moved to the supplementary section with the exception of Fig. 2a.

2] There remain a number of concerns about the phosphorylation data in western blots in several figures. In figure 2E, the data are not comparable to any other figure panels. The effects of NCOA4-RET on pMEK is very surprising (no effect) which raises concerns on the effects on ERK in subsequent figures. In figure 4b, there does not really look to be an effect of cabozantinib on ERK phosphorylation, particularly for deltaRET. If the relatively reduced levels of total RET in these lanes is taken into account it looks like the amount of pERK would not be decreased significantly. It suggests the ERK activation they are seeing might not be RET dependent in this model as it isn't affected by cabozantinib.

We agree that in order to make figures comparable, we revised figures panels to now present pMEK/MEK and p-P70 S6/P70 S6 as signaling markers. As suggested by reviewer 1 in the previous revision, we find MEK to be a more consistent marker than ERK for MAPK pathway signaling in both cell lines. In the current revised version, we replaced ERK with MEK results and as a response to comment #1, we included NIH/3T3 signaling results in the main Fig. 3b and moved MCF10A signaling results to Supplementary Fig. 2b.

Consistent with Fig. 2e where expression of NCOA4-RET and Δ RET correlate with increased phosphorylation of P70 S6 or MEK, respectively, blots in Fig. 3b for NIH/3T3 cells reveal a more effective inhibition of phosphorylation of P70 S6 in NCOA4-RET cells and inhibition of phosphorylation of MEK in Δ RET cells when treated with cabozantinib. Similar changes are noted in MCF10A cells for the respective rearrangements.

In response to the comment regarding the effect of reduced total RET levels on MAPK signaling for Supplementary Fig. 2b (previously Fig. 4b, with ERK data), we normalized MEK signaling intensity to total Δ RET levels. After normalization, we still observe a significant reduction in signaling with cabozantinib (shown in graph here) despite total Δ RET level changes. Reduction of total RET levels with alternative tyrosine kinase inhibitors (TKIs) in the setting of *RET* point mutations has been observed in previous publications, and may be attributable to indirect or an off-target effect of multikinase inhibitors¹.

1. Mologni, L., Redaelli, S., Morandi, A., Plaza-Menacho, I. & Gambacorti-Passerini, C. Ponatinib is a potent inhibitor of wild-type and drug-resistant gatekeeper mutant RET kinase. *Molecular and cellular endocrinology* **377**, 1-6 (2013).

3] The mouse tumor growth models remain hard to compare. The authors use 10 X more cells for the NCOA4RET model compared to the delta RET and WTRET, and only ½ the number of animals (Figure 5 legend). Just about all you can say about these data is that NCOA4RET is good at making tumors and the other constructs are less consistently so. The tumor sizes for delta RET are quite variable (Figure 5B) and this make the data in figure 5C difficult to rationalize.

The initial comparison of tumor growth in athymic nude mice using same cell numbers in Fig. 4a supports the reviewer comment that NCOA4-RET cells form tumors more rapidly than ΔRET cells. However, the cabozantinib mouse studies were designed to test if each individual RET altered tumor growth could be inhibited by a RET inhibitor. In order to test the drug in the same mouse background and to allow for drug treatment within the same time frame, a greater number of cells was required for injection in non-NCOA4-RET cells to achieve tumor growth and tumor size in athymic nude mice.

We have now clarified this in the results that the purpose of testing cabozantinib in mouse models was to individually assess drug response in each model (page 10, lines 214-219) and not to compare between models. Differences in the oncogenic potential between NCOA4-RET and ΔRET cells from *in vitro* phenotypes (Fig. 2) and early tumorigenic assays (Fig. 4a) guided selection of different cell numbers for these cabozantinib mouse studies. We increased sample size for the comparatively less potent ΔRET model to ensure sufficient power to make a statistically sound comparison between vehicle and treatment groups.

NCOA4-RET is clearly a more aggressive, oncogenic model in comparison with ΔRET. We would like to highlight that ΔRET, though less potent, also results in tumors with sensitivity to cabozantinib (Fig. 5b). We have included a high exposure western blot image for Fig. 5c, where, consistent with reduction in ΔRET tumor size observed in Fig. 5b, there is also a decrease in ΔRET protein levels in Fig. 5c, and concomitant clearance of ΔRET tumor cells in Fig. 5d, all verifying the inhibitory effect of cabozantinib in the ΔRET model.

4] It seems on looking through the data, that the authors have identified an oncogenic RET form, the NCOA4RET, but that the delta RET may not be a very oncogenic form, if at all. It is not a very strong promoter of proliferation or colony formation compared to the WTRET form (Fig 2B). It doesn't induce pMEK or p70S6 more strongly (Fig 2E) and is significantly less able to induce pAKT or pERK than WTRET. It is a very slow inducer of tumor growth (Fig 3A, 5B). It's response to cabozantinib is quite inconsistent (Fig 5bcd, SupFig 2C). Perhaps the authors should consider

moving the discussion of stronger and weakly transforming rearrangements up front in their manuscript and discuss the two variants from that perspective throughout. It would make the data easier to compare and would clearly highlight the differences between their models, which is an important point for discussion.

As rightly identified, differences in oncogenic potential of NCOA4-RET and Δ RET cells from *in vitro* (Fig. 2) and tumorigenic assays (Fig. 4a) is evident. Difference in oncogenic potential between variants has been previously reported for RET missense mutations and fusions in thyroid cancers and this is known to correlate with clinical phenotypes for hereditary thyroid cancer. In addition, activation by different downstream signaling partners is also reported. One can speculate that NCOA4-RET, due to the dimerizing gene partner, leads to an aggressive phenotype and a different signaling pattern whereas the truncated kinase in Δ RET results in constitutively active signaling similar to wildtype. In addition, Δ RET is modeled after RASGEF1A-RET and differences in tissue level expression between RET and RASGEF1A can further drive oncogenic potential by influencing expression levels of Δ RET, which will be under the more ubiquitous RASGEF1A promoter in the breast tissue for this fusion. Finally, co-events, similar to synergistic transformation of primary B cells by MYC, CCND1, and BCL2 may also play a role in the phenotypes observed in patient tumors with less oncogenic RET rearrangements². We now focus on clarifying this in the discussion as suggested.

2. Nakagawa, M., Tsuzuki, S., Honma, K., Taguchi, O. & Seto, M. Synergistic effect of Bcl2, Myc and Ccnd1 transforms mouse primary B cells into malignant cells. *Haematologica* **96**, 1318-1326 (2011).

Additional comments

Lines 8081 different numbers of RET alterations mentioned in these 2 lines (121 and 122)

One of the numbers represents number of independent breast cancer cases carrying RET alterations (121) and the other represents number of RET alterations (122). One case carried a RET amplification and a RET rearrangement, resulting in a total of 122 RET alterations identified in 121 cases. This is clarified on page 4, lines 80 - 84 as follows: “RET genomic alterations were observed in 1.2% (121/9,693) of independent breast cancer cases with one case harboring two RET alterations (RET amplification and RET rearrangement), resulting in a total of 122 RET genomic alterations identified. This included 16 rearrangements, 25 missense mutations, and 81 amplifications (median copy number...).”

In figure 2E The authors use pMEK and p70S6 to compare the activity of all their RET constructs, but all other figures and assays compare activity of ERK and AKT- The authors should replace this panel with comparable data to that in the other figures.

To ensure consistency with Fig. 2e, we have revised other figures by presenting p-P70 S6/P70 S6 and p-MEK/MEK signaling markers for comparisons across panels.

Reviewer #1 (Remarks to the Author):

With this second revision, Authors have addressed Reviewer comments. These new data help specifying that breast cancer can be associated to different activated RET alleles featuring different intensities of mitogenic and tumorigenic activity.

Minor point: line 340, page 15, ".....aggressive oncogenic phenotype than...." Probably: "...more aggressive oncogenic phenotype than.."

Reviewer #2 (Remarks to the Author):

The authors have revised and further improved the manuscript. They have improved the clarity of their assays and further qualified comparisons of the RETamp, deltaRET and NCOA4-RET mutant conditions.

There remain several experimental and representation issues which they should adjust to improve clarity and interpretation of the manuscript.

1] The authors were requested by both reviewers to normalize the pathways investigated downstream of RET so that data was more comparable between figures. Although they indicate that they have done this in their response to reviewers, Figure 4D still shows MEK/pMEK and AKT/pAKT rather than MEK and p70S6. Could the authors update this to show the same pathways throughout the manuscript.

2] Figure 2D- The graphs of RET phosphorylation level are normalized to the RETK758M mutant. But this is a kinase dead mutant and does not phosphorylate on Y905. Meaningful comparisons cannot be made if you normalize to zero. The authors need to chose another condition and normalize to that for example to the M918T RET condition. This would been to be clarified in the text also.

Minor point

The revised figures use a mixture of upper and lower case designations for RET, MEK etc. All human protein names should be in all capitals. The authors should look correct this in Figures 1e, 2a-e, 3b and c, 4d (antibody names).

We thank the editors and the reviewers for the opportunity to further improve the clarity of our findings. The comments throughout the revision process have significantly contributed to refining our representation and discussion of the results.

Reviewer #1 (Remarks to the Author):

With this second revision, Authors have addressed Reviewer comments. These new data help specifying that breast cancer can be associated to different activated RET alleles featuring different intensities of mitogenic and tumorigenic activity.

Minor point: line 340, page 15, "..... aggressive oncogenic phenotype than...." Probably: "...more aggressive oncogenic phenotype than.."

We are including the following sentence in response to the above comment which is now under Page 16, Lines 361 - 362: "*NCOA4-RET* potentially confers a **more** aggressive oncogenic phenotype than *RASGEF1A-RET*."

Reviewer #2 (Remarks to the Author):

The authors have revised and further improved the manuscript. They have improved the clarity of their assays and further qualified comparisons of the RETamp, deltaRET and NCOA4-RET mutant conditions.

There remain several experimental and representation issues which they should adjust to improve clarity and interpretation of the manuscript.

1] The authors were requested by both reviewers to normalize the pathways investigated downstream of RET so that data was more comparable between figures. Although they indicate that they have done this in their response to reviewers, Figure 4D still shows MEK/pMEK and AKT/pAKT rather than MEK and p70S6. Could the authors update this to show the same pathways throughout the manuscript.

We thank the reviewer for identifying this and we have now modified Fig. 4D to only include MEK and P70 S6 activation to represent MAPK and PI3K-AKT pathways respectively.

2] Figure 2D- The graphs of RET phosphorylation level are normalized to the RETK758M mutant. But this is a kinase dead mutant and does not phosphorylate on Y905. Meaningful comparisons cannot be made if you normalize to zero. The authors need to chose another condition and normalize to that for example to the M918T RET condition. This would been to be clarified in the text also.

This is an interesting insight and we now recognize that quantitative comparison to kinase dead mutant, where phosphorylation signal is assumed to be zero, is not correct. To correct this, and as suggested, we have now included comparison with the kinase active M918T mutant and also verified that it is represented accurately in the text and figure legend.

Minor point

The revised figures use a mixture of upper and lower case designations for RET, MEK etc. All human protein names should be in all capitals. The authors should look correct this in Figures 1e, 2a-e, 3b and c, 4d (antibody names).

With the latest revision, we have ensured consistent and recommended designations for protein names throughout the manuscript.